# IN-CONTEXT REINFORCEMENT LEARNING THROUGH BAYESIAN FUSION OF CONTEXT AND VALUE PRIOR

## ABSTRACT

In-context reinforcement learning (ICRL) promises fast adaptation to unseen environments without parameter updates, but current methods either cannot improve beyond the training distribution or require near-optimal data, limiting practical adoption. We introduce SPICE, a Bayesian ICRL method that learns a prior over Q-values via deep ensemble and updates this prior at test-time using in-context information through Bayesian updates. To recover from poor priors resulting from training on sub-optimal data, our online inference follows an Upper-Confidence Bound rule that favours exploration and adaptation. We prove that SPICE achieves regret-optimal behaviour in both stochastic bandits and finite-horizon MDPs, even when pretrained only on suboptimal trajectories. We validate these findings empirically across bandit and control benchmarks. SPICE achieves near-optimal decisions on unseen tasks, substantially reduces regret compared to prior ICRL and meta-RL approaches while rapidly adapting to unseen tasks and remaining robust under distribution shift.

## 1 INTRODUCTION

Following the success of transformers with in-context learning abilities Vaswani et al. (2017), In-Context Reinforcement Learning (ICRL) emerged as a promising paradigm Chen et al. (2021); Zheng et al. (2022). ICRL aims to adapt a policy to new tasks using only a context of logged interactions and no parameter updates. This approach is particularly attractive for practical deployment in domains where training classic online RL is either risky or expensive, where abundant historical logs are available, or where fast gradient-free adaptation is required. Examples include robotics, autonomous driving or buildings energy management systems. ICRL improves upon classic offline RL by amortising knowledge across tasks, as a single model is pre-trained on trajectories from many environments and then used at test time with only a small history of interactions from the test task. The model must make good decisions in new environments using this in-context dataset as the only source of information Moeini et al. (2025).

Existing ICRL approaches suffer from three main limitations. First, behaviour-policy bias from supervised training objectives: methods trained with Maximum Likelihood Estimation (MLE) on actions inherit from the same distribution as the behaviour policy. When the behaviour policy is suboptimal, the learned model performs poorly. Many ICRL methods fail to improve beyond the pretraining data distribution and essentially perform imitation learning Dong et al.; Lee et al. (2023). Second, existing methods lack uncertainty quantification and inference-time control. Successful online adaptation requires epistemic uncertainty over action values to enable temporally coherent exploration. Most ICRL methods expose logits but not actionable posteriors over Q-values, which are needed for principled exploration like Upper Confidence Bound (UCB) or Thompson Sampling (TS) Lakshminarayanan et al. (2017); Osband et al. (2016; 2018); Auer (2002); Russo et al. (2018). Third, current algorithms have unrealistic data requirements that make them unusable in most real-world deployments. Algorithm Distillation (AD) Laskin et al. (2022) requires learning traces from trained RL algorithms, while Decision Pretrained Transformers (DPT) Lee et al. (2023) needs optimal policy to label actions. Recent work has attempted to loosen these requirements, like Decision Importance Transformers (DIT) Dong et al. and In-Context Exploration with Ensembles (ICEE) Dai et al. (2024). However, these methods lack explicit measure of uncertainty and test-time controller for exploration and efficient adaptation.

To address these limitations, we introduce SPICE (**S**haping **P**olicies **I**n-**C**ontext with **E**nsemble prior), a Bayesian ICRL algorithm that maintains a prior over Q-values using a deep ensemble and updates this prior with state-weighted evidence from the context dataset. The resulting per-action posteriors can be used greedily in offline settings or with a posterior-UCB rule for online exploration, enabling test-time adaptation to unseen tasks without parameter updates. We prove that SPICE achieves regret-optimal performance in both stochastic bandits and finite-horizon MDPs, even when pretrained only on suboptimal trajectories. We test our algorithm in bandit and dark room environments to compare against prior work, demonstrating that our algorithm achieves near-optimal decision making on unseen tasks while substantially reducing regret compared to prior ICRL and meta-RL approaches. This work paves the way for real-world deployment of ICRL methods, which should feature good uncertainty quantification and test-time adaptation to new tasks without relying on unrealistic optimal control trajectories for training.

## 2 RELATED WORK

**Meta-RL.** Classical meta-reinforcement learning aims to learn to adapt across tasks with limited experience. Representative methods include $RL^2$ Duan et al. (2016), gradient-based meta-learning such as MAML Finn et al. (2017); and probabilistic context–variable methods such as PEARL Rakelly et al. (2019). These approaches typically require online interaction and task-aligned adaptation loops during deployment.

**Sequence modelling for decision-making.** Treating control as sequence modelling has proven effective with seminal works such as Decision Transformer (DT) Chen et al. (2021) and Trajectory Transformer models Janner et al. (2021). Scaling variants extend DT to many games and longer horizons Lee et al. (2022); Correia & Alexandre (2023), while Online Decision Transformer (ODT) blends offline pretraining with online fine-tuning via parameter updates Zheng et al. (2022). These works paved the way for in context decision making.

**In-context RL via supervised pretraining.** Two influential ICRL methods are Algorithm Distillation (AD) Laskin et al. (2022), which distills the learning dynamics of a base RL algorithm into a Transformer that improves in-context without gradients, and Decision-Pretrained Transformer (DPT) Lee et al. (2023), which is trained to map a query state and in-context experience to optimal actions and is theoretically connected to posterior sampling. Both rely on labels generated by strong/optimal policies (or full learning traces) and therefore inherit behaviour-policy biases from the data Moeini et al. (2025). DIT (Dong et al.) improves over behaviour cloning by reweighting a supervised policy with in-context advantage estimates, but it remains a purely supervised objective: it exposes no calibrated uncertainty, produces no per-action posterior, and lacks any inference-time controller or regret guarantees. ICEE (Dai et al., 2024) induces exploration–exploitation behaviour inside a Transformer at test time, yet it does so heuristically, without explicit Bayesian updates, calibrated posteriors, or theoretical analysis. By contrast, SPICE is the first ICRL method to (i) learn an explicit value prior with uncertainty from suboptimal data, (ii) perform Bayesian context fusion at test time to obtain per-action posteriors, and (iii) act with posterior-UCB, yielding principled exploration and a provable $O(\log K)$ regret bound with only a constant warm-start term.

## 3 SPICE: BAYESIAN IN-CONTEXT DECISION MAKING

In this section, we introduce the key components of our approach. We begin by formalising the ICRL problem and providing a high-level overview of our method in Sec. 3.1. The main elements of the model architecture and training objective are described in Sec. 3.2. Our main contribution, the test-time Bayesian fusion policy, is introduced in Sec. 3.3

### 3.1 METHOD OVERVIEW

Consider a set $\mathcal{T}$ of tasks with a state space $\mathcal{S}$, an action space $\mathcal{A}$, an horizon $H$, a per-step reward $r_t$, and discount $\gamma$. In in-context reinforcement learning, given a task $T \sim \mathcal{T}$ the agent must chose actions to maximise the expected discounted return over the trajectory. During training, the agent learns from trajectories collected either offline or online on different tasks. A test time the agent is

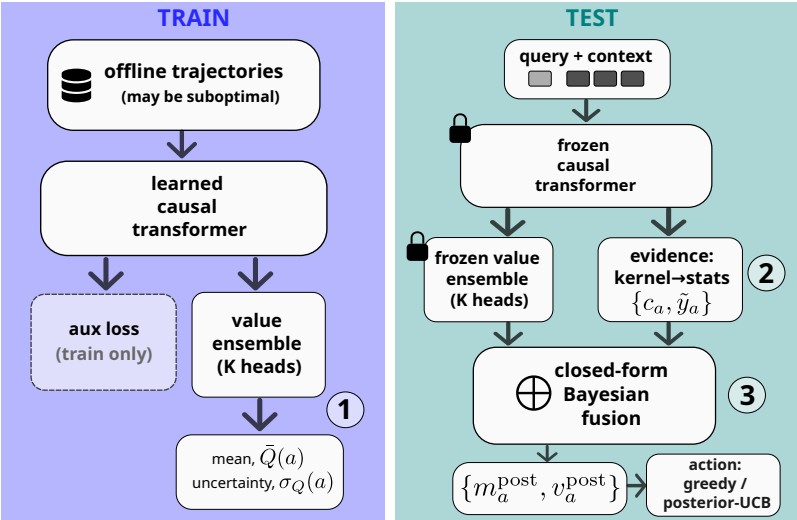

Figure 1: **Training and Test-Time Overview.** SPICE learns a causal-transformer backbone and a K-head value ensemble from offline trajectories, then performs test-time adaptation without gradients by combining the ensemble's value prior with context-derived evidence via a closed-form Bayesian update. Circled numbers mark core contributions: ① ensemble prior with calibrated uncertainty, ② kernel-based evidence extraction from multi-episode context, and ③ closed-form Bayesian fusion enabling greedy (offline) / posterior-UCB (online) action selection.

given a new task and a context $C = \{(s_t, a_t, r_t, s_{t+1})\}_{t=1}^{h}$ and a query state $s_{qry}$. The context either comes from offline data or collected online. The goal is to choose an action $a = \pi(s_{qry}, C)$ that maximises the expected return. The adaptation of the policy the new task is done in context, without any parameter update.

Our algorithm, Shaping Policies In-Context with Ensemble prior (SPICE), solves the ICLR problem with a Bayesian approach. It combines a value prior learned from training tasks with task-specific evidence extracted from the test-time context. SPICE first encodes the query and context using a transformer trunk and then produces a calibrated per-action value prior via a deep ensemble. Weighted statistics are extracted from the context using a kernel that measures state similarity. Prior and context evidence are then fused through a closed-form Bayesian update. Actions can be selected greedily or with respect to a posterior-UCB rule for principled exploration. This design enables SPICE to adapt quickly to new tasks and overcome the behaviour-policy bias, even when trained on suboptimal data.

SPICE introduces three key contributions: (1) a value-ensemble prior that provides calibrated epistemic uncertainty from suboptimal data, (2) a weighted representation-shaping objective that enables the trunk to support reliable value estimation, and (3) a test-time Bayesian fusion controller that produces per-action posteriors and enables coherent in-context exploration via posterior-UCB. The approach is summarised in Fig. 1 and the full algorithm is described in Algo. 1 along with the detailed architecture in Fig. 5. Note that our approach focuses on discrete action spaces $\mathcal{A}$, but it extends naturally to continuous actions. [1]

### 3.2 LEARNING THE VALUE PRIOR AND REPRESENTATION

**Transformer Trunk (sequence encoder)** Following prior work (Lee et al., 2023), a causal GPT-2 transformer is used to encode sequences of transitions. Each transition is embedded using a single linear layer $\mathbf{h}_t = \text{Linear}([s_t, a_t, s'_t, r_t]) \in \mathbb{R}^D$, where $D$ is the hidden size dimension. A

---

[1]For continuous action settings, one can replace the categorical policy head with a parametric density (e.g., Gaussian), concatenate raw action vectors instead of one-hot encodings in the value ensemble, and perform posterior updates using kernel-weighted statistics in action space.

sequence is processed as

$$([\underbrace{s_{\mathrm{qry}},\ 0,\ 0,\ 0}_{\text{query token}}],\ \underbrace{[s_1, a_1, s'_1, r_1],\dots,[s_H, a_H, s'_H, r_H]}_{\text{context transitions}})$$

and the transformer outputs a hidden vector at each position. Two decoder heads are used a policy head $\pi_\theta(a \mid \cdot)$ and a value ensemble head $Q_{\phi_k}(a \mid \cdot)$ for $k = 1, \dots, K$. The trunk maps the query and context to a shared representation; the policy head provides a training-only signal that shapes this representation. The value ensemble uses it to produce the test-time value prior.

**Value Ensemble Prior**  We attach $K$ independent value heads with randomised priors and a small anchor penalty to encourage diversity and calibration (Lakshminarayanan et al., 2017; Osband et al., 2018; Pearce et al., 2018; Wilson & Izmailov, 2020; Fort et al., 2019).

The ensemble mean and standard deviation are used as calibrated value prior for ICRL:

$$\bar{Q}(a) \ = \ \tfrac{1}{K} \sum_{k=1}^{K} Q_{\phi_k}(a), \qquad \sigma_Q(a) \ = \ \sqrt{\tfrac{1}{K-1} \sum_{k=1}^{K} \big(Q_{\phi_k}(a) - \bar{Q}(a)\big)^2}. \tag{1}$$

We treat $\bar{Q}(a)$ and $\sigma_Q^2(a)$ as the mean and variance of a Gaussian prior over $Q(a)$. The anchor penalty contributes the $\mathcal{L}_{\mathrm{anchor}}$ term to the total loss, which is an $L_2$ regularisation on the value head weights to enforce diversity and improve uncertainty calibration. Architectural details (randomised priors, anchor loss) are in Appendix A.

**Representation Shaping with Weighted Supervision**  Although the policy head is not used at test time, its weighted supervision shapes the trunk so that the value ensemble receives de-biased, reward-relevant, and uncertainty-aware features (correcting behaviour-policy bias, upweighting high-advantage examples, and focusing on epistemically uncertain regions). This improves the value estimation, especially when the training label $a_b^\star$ is suboptimal. The policy loss is calculated as the expected weighted cross-entropy over a batch of training examples $b$, where $h_{b,t}$ is the hidden state at time $t$ for example $b$ and $a_b^\star$ is the label:

$$\mathcal{L}_\pi \ = \ \mathbb{E}_b\left[ \frac{1}{H} \sum_{t=1}^{H} \omega_b \left( -\log \pi_\theta(a_b^\star \mid \mathbf{h}_{b,t}) \right) \right], \qquad \omega_b = \omega_{\mathrm{IS}} \cdot \omega_{\mathrm{adv}} \cdot \omega_{\mathrm{epi}}. \tag{2}$$

The multiplicative weight $\omega_b$ is the product of the importance, advantage and epistemic weight factors described in Appendix A.1.

**Value Head Training**  The value ensemble is trained via TD($n$) regression and augmented by a Bayesian shrinkage loss ($\mathcal{L}_Q = \mathcal{L}_{\mathrm{TD}} + \mathcal{L}_{\mathrm{shrink}}$). This shrinkage stabilises the value estimates and acts like a per-action prior that prevents the ensemble from overfitting to sparse or noisy data. Full targets and losses are provided in Appendix A.1.

**Training Objective**  The full training loss $\mathcal{L} = \mathcal{L}_\pi + \lambda_Q \mathcal{L}_Q + \lambda_{\mathrm{anchor}} \mathcal{L}_{\mathrm{anchor}}$ is optimised using AdamW (Loshchilov & Hutter, 2019).

### 3.3 TEST-TIME BAYESIAN FUSION OF CONTEXT AND VALUE PRIOR

This section presents the key component of our algorithm: a test-time controller that combines information from the ensemble prior and context, following a UCB principle for action selection. The posterior-UCB rule turns the value uncertainty into directed exploration, allowing SPICE to adapt online even under suboptimal or biased pretraining data, where implicit in-context adaptation typically fails.

At the query state $s$ we form an action-wise posterior by combining the ensemble prior $(\bar{Q}, \sigma_Q)$ with state-weighted statistics extracted from the context. Let $w_t(s) \in [0, 1]$ denote a kernel weight that

measures how similar context state $s_t$ is to the query $s$. Instances of such kernels are uniform, cosine or RBF kernels Cleveland & Devlin (1988); Watson (1964). The performance of the Bayesian fusion critically depends on the state-similarity kernel, as a mismatch the kernel's similarity metric and the true Q function structure can corrupt the Bayesian update. To mitigate this, SPICE applies the kernel to the feature vector produced by the Transformer trunk, $h_{qry}$, not the raw state space $s$. This increases robustness, as the transformer is trained to map states with similar action values, advantage estimates and epistemic uncertainty into nearby regions in the latent space, see Section 3.2. In practice, for structured MDP state spaces like the Darkroom, we use an RBF kernel applied to the latent features $h$:

$$w_t(s) = \exp\left(-\frac{||h_{qry} - h_t||_2^2}{2\tau^2}\right) \tag{3}$$

where $h_{qry}$ is the feature vector for the query state $s_{qry}$ and $h_t$ is for the context state $s_t$. For simple, unstructured environments like the bandits, the Uniform kernel (equivalent to $\tau \to \infty$ or using a fixed count $c_a$ with all $w_t(s) \in \{0, 1\}$) is sufficient. We provide guidance for the kernel selection in new domains in Appendix A.7.

For each action, the state-weighted counts and targets are

$$c_a(s) = \sum_t w_t(s)\,\mathbb{1}[a_t = a], \qquad \tilde{y}_a(s) = \frac{\sum_t w_t(s)\,\mathbb{1}[a_t = a]\,y_t}{\max\left(1, c_a(s)\right)}. \tag{4}$$

The target $y_t$ can be chosen as immediate reward or an $n$-step bootstrapped return :

$$y_t^{(n)} = \sum_{i=0}^{n-1} \gamma^i r_{t+i} + \gamma^n \max_{a'} \bar{Q}(s_{t+n}, a'). \tag{5}$$

Given Eq. 1, a choice of kernel, and the weighted evidence $\left(c_a(s), \tilde{y}_a(s)\right)$, SPICE composes a conjugate-style posterior per action by precision additivity Murphy (2007; 2012):

**Step 1: Prior from ensemble.**

The ensemble's predictive mean and uncertainty at the query provide a Gaussian prior over $Q(a)$. The likelihood variance $\sigma^2$ specifies the noise level that we assume for the targets.

$$\mu_a^{\text{pri}} = \bar{Q}(a), \quad v_a^{\text{pri}} = \max\{\sigma_Q(a)^2, v_{\min}\}, \qquad \text{likelihood variance: } \sigma^2 \tag{6}$$

**Step 2: Precision additivity with Normal-Normal conjugacy.** We assume that $Q(a)$ follows a Normal prior $Q(a) \sim \mathcal{N}(\mu_a^{pri}, v_a^{pri})$ and that the observed kernel-weighted targets $\tilde{y}_a(s)$ are noisy samples with variance $\sigma^2/c_a(s)$. The Gaussian likelihood is $p(\tilde{y}_a(s)\,|\,Q(a)) \propto \exp\left(-\frac{c_a(s)}{2\sigma^2}(Q(a) - \tilde{y}_a(s))^2\right)$. Multiplying the prior and likelihood gives a Gaussian posterior whose precision is the sum of prior and data precisions:

$$\text{posterior:} \quad v_a^{\text{post}} = \left(\frac{1}{v_a^{\text{pri}}} + \frac{c_a(s)}{\sigma^2}\right)^{-1}, \qquad m_a^{\text{post}} = v_a^{\text{post}}\left(\frac{\mu_a^{\text{pri}}}{v_a^{\text{pri}}} + \frac{c_a(s)\,\tilde{y}_a(s)}{\sigma^2}\right). \tag{7}$$

The posterior is derived using the classical equations for Gaussian conjugate updating from Murphy (2007; 2012), a derivation can be found in Appendix A.1.

**Step 3: Action selection.** Based on this posterior distribution, we propose the following action selection:

- **Online**, the policy follows a posterior-UCB rule with exploration parameter $\beta_{\text{ucb}} > 0$ Auer (2002), allowing exploration and adaptation to the task:
$$a^\star = \arg\max_a \left(m_a^{\text{post}} + \beta_{\text{ucb}}\sqrt{v_a^{\text{post}}}\right). \tag{8}$$

- **Offline**, the policy act greedily: $a^\star = \arg\max_a m_a^{\text{post}}$.

Intuitively, the posterior mean $m_a^{post}$ aggregates prior knowledge and local context evidence, while the variance $v_a^{post}$ quantifies the remaining uncertainty. The UCB rule acts optimistically when uncertainty is large, guaranteeing efficient exploration and provably logarithmic regret bound, see Section 4. Hyperparameter choices are listed in Appendix A.8 and the pseudocode for Bayesian fusion appears in Algorithm 1 (Appendix A.1).

# 4    REGRET BOUND OF THE SPICE ALGORITHM

A key component of SPICE is the use of a posterior-UCB rule at inference time that leverages both ensemble prior and in-context data. Importantly, we show in this section that the resulting online controller achieves optimal logarithmic regret despite being pretrained on sub-optimal data. Any prior miscalibration from pretraining manifests only as a constant warm-start term without affecting the asymptotic convergence rate. We establish this result formally in both the bandit and MDP settings and provide empirical validation across bandit and MDP problems in the next section.

## 4.1    BANDIT SETTING

Consider a stochastic $A$-armed bandit setting with unknown means $\{\mu_a\}_{a=1}^A \subset \mathbb{R}$. At each round $t \in 1, ..., K$ the algorithm chooses $a_t$ and receives a reward $r_t = \mu_{a_t} + \varepsilon_t$, where $(\varepsilon_t)_{t \geq 1}$ are independent mean zero $\sigma$−sub-Gaussian noise variables. The best-arm mean is defined as $\mu_\star = \max_{a \in [A]} \mu_a$ and the gap of arm $a$ as $\Delta_a = \mu_\star - \mu_a$.

Without loss of generality, we scale rewards so that means satisfy $\mu_a \in [0.1]$ for all $a \in [A]$. Hence $0 \leq \mu_\star - \mu_a \leq 1$ and the per-round regret is at most 1. Assuming that each reward distribution is $\sigma^2$-sub-Gaussian, a current assumption in bandit analysis(Whitehouse et al., 2023; Han et al., 2024), one can derive the following tail bound for any arm $a$ and round $t \geq 1$ with $n_{a,t}$ pulls and empirical mean $\widehat{\mu}_{a,t}$ for all $\varepsilon > 0$

$$\Pr\left(\left|\widehat{\mu}_{a,t} - \mu_a\right| > \varepsilon\right) \leq 2\exp\left(-\frac{n_{a,t}\varepsilon^2}{2\sigma^2}\right) \tag{9}$$

By setting $\varepsilon = \sigma\sqrt{\frac{2\log t}{n_{a,t}}}$, one can show that with probability at least $1 - O(\frac{1}{t^2})$

$$\left|\widehat{\mu}_{a,t} - \mu_a\right| \leq \sigma\sqrt{\frac{2\log t}{n_{a,t}}}, \tag{10}$$

i.e the deviation of the empirical mean from the true mean is bounded by $\sigma\sqrt{2\log t/n_{a,t}}$ with high probability (Hoeffding's inequality; see (Hoeffding, 1963; Boucheron & Thomas, 2012)).

**Definition 1** (SPICE posterior). *Let the ensemble prior for arm $a$ be Gaussian with mean $\mu_a^{pri}$ and variance $v_a^{pri} > 0$, estimated from the value ensemble at the query (see Section 3.3). The prior pseudo-count is defined as*

$$N_a^{pri} := \frac{\sigma^2}{v_a^{pri}}, \quad \implies \quad m_{a,t}^{post} = \frac{N_a^{pri}\mu_a^{pri} + n_{a,t}\widehat{\mu}_{a,t}}{N_a^{pri} + n_{a,t}}, \quad v_{a,t}^{post} = \frac{\sigma^2}{N_a^{pri} + n_{a,t}} \tag{11}$$

*where $n_{a,t}$ and $\widehat{\mu}_{a,t}$ are the number of pulls and the empirical mean of arm $a$ up to round $t$ (these updates follow Normal-Normal conjugacy; see Murphy, 2007; 2012.) .*

**Definition 2** (SPICE inference). *SPICE acts using a posterior-UCB rule at inference time*

$$a_t \in \arg\max_{a \in \mathcal{A}}\left\{m_{a,t-1}^{post} + \beta_t\sqrt{v_{a,t-1}^{post}}\right\}, \quad \beta_t = \sqrt{2\log t} \tag{12}$$

*The schedule $\beta_t = \sqrt{2\log t}$ mirrors the classical UCB1 analysis (Auer et al., 2002).*

We now derive a regret bound for SPICE inference-time controller. The proof is given in Sec. B.

**Theorem 1** (SPICE's Regret-optimality with warm start in Bandits.). *Under the assumption of $\sigma^2$-sub-Gaussian reward distributions, the SPICE inference controller satisfies*

$$\mathbb{E}\left[\sum_{t=1}^K (\mu_\star - \mu_{a_t})\right] \leq \sum_{a \neq \star}\left(\frac{32\sigma^2\log K}{\Delta_a^2} + 4N_a^{pri}\left|\mu_a^{pri} - \mu_a\right|\right) + O(1). \tag{13}$$

Thus the cumulative regret of SPICE has an optimal logarithm rate in $K$ and any sub-optimal pre-training results only in a constant warm-start term $\sum_{a \neq \star} 4N_a^{pri}\left|\mu_a^{pri} - \mu_a\right|$ that does not scale with $K$. The leading $O(\log K)$ term matches the classical UCB1 proof (Auer et al., 2002). The additive

warm-start term depends on the prior pseudo-count $N_a^{\text{pri}} = \sigma^2/v_a^{\text{pri}}$, which behaves as prior data in a Bayesian sense (Gelman et al., 1995).

This theorem yields the following corollaries highlighting the impact of the prior quality on the regret bound.

**Corollary 1** (Bound of well-calibrated priors). *If the ensemble prior is perfectly calibrated, then $\mu_a^{pri} = \mu_a$ for all arms $a$ and the warm-start term vanishes. SPICE then reduces to classical UCB*

$$\mathbb{E}[R_K] \leq \sum_{a \neq \star} \frac{32\sigma^2 \log K}{\Delta_a^2} + O(1). \tag{14}$$

**Corollary 2** (Bound on weak priors). *If the ensemble prior has infinite variance, $v_a^{pri} \to \infty$ and therefore $N_A^{pri} \to 0$. The warm-start term vanishes and SPICE reduces to classical UCB Eq. 14.*

The regret bound shows that SPICE inherits the optimal $O(\log K)$ rate of UCB while adding a constant warm-start cost from pretraining. The posterior mean in Eq. 11 is a convex combination of the empirical and prior means and the variance is shrinking at least as fast as $O(1/n_{a,t})$. Early decisions are influenced by the prior, but as $n_{a,t}$ grows, the bias term vanishes and learning relies entirely on observed rewards. A miscalibrated confident prior increases the warm-start constant but does not affect asymptotics, a well-calibrated prior eliminates the warm-start entirely and an uninformative prior ($v_a^{\text{pri}} \to \infty$) reduces SPICE to classical UCB. In practice, this means that SPICE can exploit structure from suboptimal pretraining when it is useful, while remaining safe in the long run, as its regret matches UCB regardless of the prior quality.

## 4.2 EXTENSION TO MARKOV DECISION PROCESSES

We extend our inference-time analysis from stochastic bandits to finite-horizon Markov Decision Processes (MDPs). We show that SPICE achieves the minimax-optimal regret rate for finite-horizon MDPs (Auer et al., 2008; Azar et al., 2017), while any miscalibration in the ensemble prior contributes only a constant warm-start term, exactly mirroring the bandit case.

We consider a finite-horizon MDP $M = \langle \mathcal{S}, \mathcal{A}, P, R, H \rangle$ with finite state and action spaces $|\mathcal{S}| = S$, $|\mathcal{A}| = A$, transition kernel $P$, reward function $R$ bounded in $[0,1]$, and fixed episode length $H$. We run SPICE for $K$ episodes, with initial state $s_1^k$ in episode $k$, and write $T := KH$ for the total number of interaction steps. Let $Q_\star$ and $V_\star$ denote the optimal $Q$-function and value function, and let $\pi_k$ be the policy used in episode $k$ by the SPICE controller. The cumulative regret is

$$\mathbb{E}[\text{Regret}_K] := \mathbb{E}\left[\sum_{k=1}^{K} \left(V_\star(s_1^k) - V_{\pi_k}(s_1^k)\right)\right].$$

**Definition 3** (MDP posterior and TD-based evidence). *For each state-action pair $(s,a)$, SPICE maintains a Gaussian prior*

$$Q(s,a) \sim \mathcal{N}(\mu_{s,a}^{\text{pri}}, v_{s,a}^{\text{pri}}),$$

*with prior pseudo-count $N_{s,a}^{\text{pri}} := \sigma_Q^2/v_{s,a}^{\text{pri}}$, in analogy to the bandit setting (Definition 1). When $(s_t, a_t) = (s,a)$ is visited at time $t$, SPICE constructs an $n$-step TD target*

$$y_t^{(n)} = \sum_{i=0}^{n-1} \gamma^i r_{t+i} + \gamma^n \max_{a'} \overline{Q}(s_{t+n}, a'),$$

*where $\overline{Q}$ is the ensemble estimate and $\gamma \in [0,1]$ (for episodic finite-horizon problems one may take $\gamma = 1$). Let $\tilde{y}_{s,a,t}$ denote a kernel-weighted average of such targets collected for $(s,a)$ up to time $t$. The SPICE posterior for $(s,a)$ is obtained by combining the Gaussian prior with a Gaussian likelihood on $\tilde{y}_{s,a,t}$ with variance proxy $\sigma_Q^2$, using the same precision-additivity rule as in equation 11, yielding posterior mean $m_{s,a,t}^{\text{post}}$ and variance $v_{s,a,t}^{\text{post}}$.*

We impose the following assumption on the TD-based evidence, which matches the conditions used in our regret bound.

**Assumption 1** (TD evidence quality). *For every $(s,a)$ there exists an $n$ such that, for the $n$-step TD targets $y_t^{(n)}$ defined above and history $\mathcal{F}_{t-1}$ up to time $t-1$,*

$$\mathbb{E}\big[y_t^{(n)} \mid s_t = s, a_t = a, \mathcal{F}_{t-1}\big] = Q_\star(s,a), \qquad y_t^{(n)} - Q_\star(s,a) \text{ is conditionally } \sigma_Q\text{-sub-Gaussian},$$

*for some variance proxy $\sigma_Q^2 \le c_H H$ depending only on the horizon $H$.*

**Theorem 2** (SPICE's Regret-optimality in Finite-Horizon MDPs). *Consider a finite-horizon MDP $M = \langle \mathcal{S}, \mathcal{A}, P, R, H \rangle$ satisfying the conditions above and Assumption 1. Let the SPICE inference controller maintain for each $(s,a)$ a Gaussian prior $Q(s,a) \sim \mathcal{N}(\mu_{s,a}^{\mathrm{pri}}, v_{s,a}^{\mathrm{pri}})$ and act with the posterior-UCB rule*

$$a_t \in \arg\max_{a \in \mathcal{A}} \big\{ m_{s_t,a,t}^{\mathrm{post}} + \beta_t \sqrt{v_{s_t,a,t}^{\mathrm{post}}} \big\}.$$

*Let $N_{s,a}^{\mathrm{pri}} := \sigma_Q^2 / v_{s,a}^{\mathrm{pri}}$ be the prior pseudo-count and denote $N^{\max} := \max_{s,a} N_{s,a}^{\mathrm{pri}}$. Assume an exploration schedule of the form*

$$\beta_t := C_\beta \sqrt{\log(SAT)}, \qquad C_\beta \ge 2\sqrt{1 + N^{\max}},$$

*which is of order $\Theta(\sqrt{\log T})$ and whose constant depends only on the prior. Then the cumulative regret over $K$ episodes satisfies*

$$\mathbb{E}[\mathrm{Regret}_K] \le O\big(H\sqrt{SAK}\big) + \sum_{(s,a) \in \mathcal{S} \times \mathcal{A}} O\Big(N_{s,a}^{\mathrm{pri}} |\mu_{s,a}^{\mathrm{pri}} - Q_\star(s,a)|\Big), \tag{15}$$

*where $\pi_k$ is the policy used in episode $k$ and the constants in the big-O notation do not depend on $K$.*

Thus SPICE attains the optimal asymptotic regret rate $O(H\sqrt{SAK})$ for finite-horizon MDPs (Auer et al., 2008). As in the bandit setting, the only effect of suboptimal pretraining is a constant warm-start cost

$$\sum_{(s,a)} O\Big(N_{s,a}^{\mathrm{pri}} |\mu_{s,a}^{\mathrm{pri}} - Q_\star(s,a)|\Big),$$

which does not grow with $K$. If the ensemble prior is perfectly calibrated, this term vanishes; if it is weak (large variance, small $N_{s,a}^{\mathrm{pri}}$), SPICE essentially reduces to a standard optimistic value-iteration-style controller with optimal regret guarantees.

# 5 LEARNING IN BANDITS

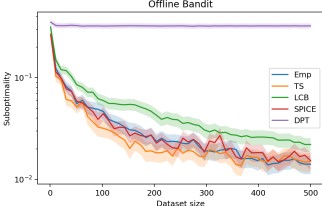 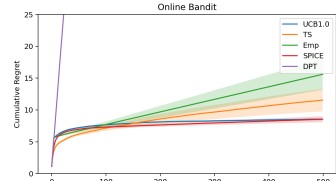 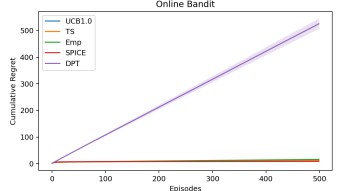

(a) Offline: suboptimality vs. context size $h$. Lower is better.

(b) Online: cumulative regret (zoomed). Lower is better.

(c) Online: cumulative regret (full scale).

Figure 2: **Bandit performance evaluation.** **(a)** Offline selection quality. **(b)** Online cumulative regret (zoomed view). **(c)** Online cumulative regret (full scale). Shaded regions are $\pm$ SEM over $N{=}200$ test environments.

We test our algorithm using the DPT evaluation protocol (Lee et al., 2023). Each task is a stochastic $A$-armed bandit with Gaussian rewards. Unless noted, $A{=}5$ and horizon $H{=}500$. The pretraining is intentionally heterogeneous: for each training task we sample a behaviour distribution $p = (1 - \omega)\,\mathrm{Dirichlet}(\mathbf{1}) + \omega\,\delta_{i_\star}$ over arms (with the label $i_\star$ being a random arm), resulting in random-policy contexts with uneven coverage. To quantify the sensitivity to the data quality, Appendix D.2 tackles a less-poor setting with $80\%$ optimal labels and the same mixed behaviour in the pretraining dataset. Further details are given in Appendix D and Appendix A.8.

Results are presented in Fig. 2 and Fig. 3. **Offline**, SPICE and TS achieve the lowest suboptimality across $h$, while LCB is competitive early but remains above TS/SPICE. DPT is flat and far from optimal in this weak-data regime. **Online**, SPICE attains the lowest cumulative regret among learned methods and tracks the classical UCB closely (Fig. 2b and Fig. 2c). Under increasing reward noise, SPICE, TS, UCB, and Emp degrade smoothly with small absolute changes, whereas DPT's final regret remains two orders of magnitude larger, indicating failure to adapt from weak logs (Fig. 3).

SPICE achieves logarithmic online regret from sub-optimal pretraining. Its posterior-UCB controller inherits $O(\log H)$ regret, with any prior miscalibration

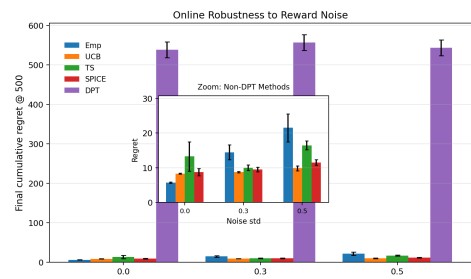

Figure 3: **Robustness to reward noise.** Final regret at $H{=}500$ for different noise levels ($\sigma \in \{0.0, 0.3, 0.5\}$). Bars are $\pm$ SEM over $N{=}200$ test environments.

contributing only a constant warm-start term; the empirical curves match this prediction. Even with non-optimal pretraining, Bayesian fusion quickly overrides prior bias as evidence accrues, while DPT remains tied to its supervised labels.

## 6 LEARNING IN MARKOV DECISION PROCESSES

The Darkroom is a $10{\times}10$ gridworld with $A{=}5$ discrete actions and a sparse reward of $1$ only at the goal cell. We pretrain on 100,000 environments using trajectories from a uniform behaviour policy and the "weak-last" label (the last action in the context), which provides explicitly suboptimal supervision. This is an intentionally worst-case dataset: roll-ins are uniform (random policy) and labels are chosen to be the last action in the context, so the prior must be learned from rewards rather than imitation. The evaluation is a test to extrapolate to out-of-distribution goals and represents a fundamental shift in the reward function $R$. Testing uses $N{=}100$ held-out goals, horizon $H{=}100$, and identical evaluation for all methods. Further details are given in Appendix D and Appendix A.8.

Under weak supervision, DPT = AD-BC, as DPT is trained by cross-entropy to predict a single action label from the [query; context] sequence. With the "weak-last" dataset this label is simply the last action taken by a uniform behaviour policy. Algorithm Distillation (AD) with a behaviour-cloning teacher (AD-BC) optimises the same loss on the same targets, so both reduce to contextual behaviour cloning on suboptimal labels. Lacking reward-aware targets or calibrated uncertainty, the resulting policy remains bound to the behaviour and fails to adapt online, hence the flat returns and near-linear regret. In this environment, SPICE adapts quickly and achieves high return with a regret curve that flattens after a short warm-up (Figs. 4a–4b). DPT, identical to AD-BC in this regime, exhibits near-linear regret and essentially zero return. We include PPO as a single-task RL reference for sample-efficiency; it improves but remains far below SPICE.

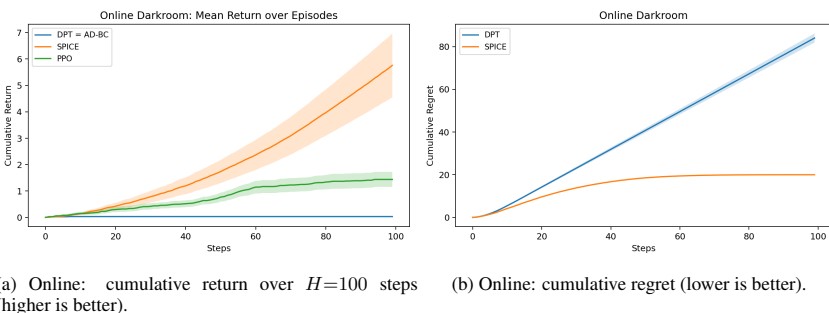

(a) Online: cumulative return over $H{=}100$ steps (higher is better).

(b) Online: cumulative regret (lower is better).

Figure 4: **Darkroom (MDP) results.** Models are pretrained on uniformly collected, weak-last labeled trajectories and evaluated online on $N{=}100$ held-out tasks for $H{=}100$ steps. Shaded regions denote $\pm$ SEM across tasks. This setup is intentionally worst-case: contexts come from a uniform random policy and labels are uninformative.

## 7 DISCUSSION

SPICE addresses limitations of current ICRL methods using minimal changes to the sequence-modelling recipe: a lightweight value ensemble is attached to a shared transformer and learns the value prior at the query state; the transformer trunk is learned using a weighted loss to shape better representations feeding into the value ensemble; at inference, the value ensemble prior is fused with state-weighted statistics extracted from the provided context of the test task, resulting in per-action posteriors that can be used greedily offline or with a posterior-UCB rule for principled exploration online. SPICE is designed to learn a good-enough structural prior from the suboptimal data to leverage knowledge such as reward sparsity and consistent action effects across different environments. The value ensemble provides calibrated uncertainty that behaves as if the prior contributed a small number of virtual samples: it influences the posterior in the first few steps but is quickly outweighed as more data from the test environment is collected. This equips SPICE with two advantages: a strong warm start from weak data and principles posterior-UCB exploration, enabling rapid adaptation to new tasks and low regret in practice.

Theoretically, we show that SPICE achieves optimal $O(\log K)$ regret in stochastic bandits and the optimal $O(H\sqrt{SAK})$ regret rate in finite-horizon MDPs, with any pretraining miscalibration contributing only to a constant warm-start term. We validate this empirically, demonstrating that SPICE achieves logarithmic regret when trained on suboptimal data, while sequence-only ICRL baselines achieve lower return and linear regret (Fig. 4). Similarly, SPICE performs nearly optimal in offline selection on held-out tasks in weak data regimes, a setting where classic ICRL perform extremely poorly (Fig. 2a).

Future work will address some of SPICE's limitations. SPICE uses kernel-weighted counts to extrapolated state proximity at inference. The kernel choice can be important in highly non-stationary or partially observable settings, where poorly chosen kernels can either over-fit or over-smooth context evidence. Additionally, SPICE assumes that the ensemble produces reasonably calibrated priors. If the prior is systematically misspecified, the posterior fusion may inherit its bias. This can slow early adaptation despite the regret guarantees.

## 8 CONCLUSION

We introduce SPICE, a Bayesian in-context reinforcement learning method that i) learns a value ensemble prior from suboptimal data via TD($n$) regression and Bayesian shrinkage, ii) performs Bayesian context fusion at test time to obtain per-action posteriors and iii) acts with a posterior-UCB controller, performing principled exploration. The design is simple: attach lightweight value heads to a Transformer trunk and keep adaptation entirely gradient-free. SPICE addresses two persistent challenges in ICRL: behaviour-policy bias during pretraining and the lack of calibrated value uncertainty at inference. Theoretically, we show that the SPICE controller has optimal logarithmic regret in stochastic bandits and optimal $O(H\sqrt{SAK})$ regret in finite-horizon MDPs, any pretraining miscalibration contributes only to a constant warm-start term. Empirical results show that SPICE achieves near-optimal offline decisions and online regret under distribution shift on bandits and control tasks.

## 9 REPRODUCIBILITY STATEMENT

We provide the details needed to reproduce all results. Algorithmic steps and test-time inference are given in Appendix A.1; model architecture, losses, and all hyperparameters are listed in Appendix A; data generators, evaluation protocols, and an ablation study are specified in Appendix D, with metrics, horizons, and noise levels matched to the DPT protocol Lee et al. (2023). Figures report means $\pm$ s.e.m. over the stated number of tasks and seeds, and we fix random seeds for every run. We use only standard benchmarks and public baselines; no external or proprietary data are required. We will release code, configuration files, and checkpoints upon publication to facilitate exact replication.

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

APPENDIX

# A  IMPLEMENTATION AND EXPERIMENTAL DETAILS

## A.1  SPICE ALGORITHM

---

**Algorithm 1** SPICE: Training and Test-Time Bayesian Fusion

---

0: **Inputs:** ensemble size $K$, prior scale $\alpha$, horizon $H$, discount $\gamma$, TD($n$) length $n$, kernel $(\phi, \tau)$, noise variance $\sigma^2$, prior-variance floor $v_{\min}$

0: **Model:** GPT-2 trunk; policy head $\pi_\theta$; value heads $Q_{\phi_k} = f_k + \alpha\, p_k$ with frozen priors $p_k$

0: **Training loop (contexts):**

0: **for** batch $\{(s_t, a_t, r_t, s_{t+1})_{t=1}^H, a^\star\}$ **do**

0:     Encode *[query; context]* with the transformer

0:     Obtain logits $\pi_\theta$, ensemble values $Q_{\phi_{1:K}}$; define $\bar{Q}, \sigma_Q$

0:     Compute weights $\omega = \omega_{\text{IS}} \cdot \omega_{\text{adv}} \cdot \omega_{\text{epi}}$

0:     Update policy with weighted cross-entropy $\mathcal{L}_\pi$

0:     Update value heads with TD($n$) regression + conjugate shrinkage + anchor regulariser

0: **end for**

0: **Test-time decision (query state $s$ with context $\mathcal{C}$):**

0: Run transformer to get prior $(\bar{Q}(a), \sigma_Q(a))$

0: Form state-weighted evidence $(c_a(s), \tilde{y}_a(s))$ via kernel weights

0: Fuse prior and evidence by precision additivity to get posterior $(m_a^{\text{post}}, v_a^{\text{post}})$

0: Select action $a^\star = \arg\max_a \left( m_a^{\text{post}} + \beta_t \sqrt{v_a^{\text{post}}} \right)$ (UCB) or $\arg\max_a m_a^{\text{post}}$ (greedy) =0

---

## A.2  WEIGHTED OBJECTIVES FOR REPRESENTATION SHAPING

The policy loss is calculated as the expected weighted cross-entropy over a batch of training examples $b$, where $h_{b,t}$ is the hidden state at time $t$ for example $b$ and $a_b^\star$ is the label:

$$\mathcal{L}_\pi = \mathbb{E}_b\left[ \frac{1}{H} \sum_{t=1}^H \omega_b \left( -\log \pi_\theta(a_b^\star \mid \mathbf{h}_{b,t}) \right) \right], \qquad \omega_b = \omega_{\text{IS}} \cdot \omega_{\text{adv}} \cdot \omega_{\text{epi}}. \tag{16}$$

The multiplicative weight $\omega_b$ is the product of three weight factors described below.

**(i) Propensity correction.**  Offline datasets reflect the action selection of the behaviour policy $\pi_b(\cdot \mid s)$, which induces a mismatch between the supervised training target and the uniform reference action distribution. To remove this behaviour-policy bias and recover the target likelihood under a uniform distribution $\pi_u(\cdot \mid s)$, labeled samples can be re-weighted with an importance ratio (Dai et al., 2024):

$$\omega_{\text{IS}} = \text{clip}\left( \frac{\pi_u(a_b^\star \mid s)}{\pi_b(a_b^\star \mid s)} \,, \, 0, \, c_{\text{iw}} \right), \qquad \pi_u(a \mid s) = \frac{1}{|\mathcal{A}|}. \tag{17}$$

Intuitively, overrepresented actions under $\pi_b$ are downweighted, and rare but informative actions are upweighted.

**(ii) Advantage weighting.**  Inspired by (Wang et al., 2018; Dai et al., 2024; Peng et al., 2019), we upweight transitions whose estimated advantage is positive so that the trunk allocates more capacity to reward-relevant behaviours, thereby improving learning from suboptimal data. The advantage is estimated using the Q-value ensemble:

$$\omega_{\text{adv}} = \text{clip}\left( \exp\left( \frac{A(s, a_b^\star)}{\tau_{\text{adv}}} \right), \varepsilon, c_{\text{adv}} \right), \quad A(s, a) := \left( \frac{1}{K} \sum_k Q_{\phi_k}(s, a) \right) - \frac{1}{|\mathcal{A}|} \sum_{a'} \left( \frac{1}{K} \sum_k Q_{\phi_k}(s, a') \right). \tag{18}$$

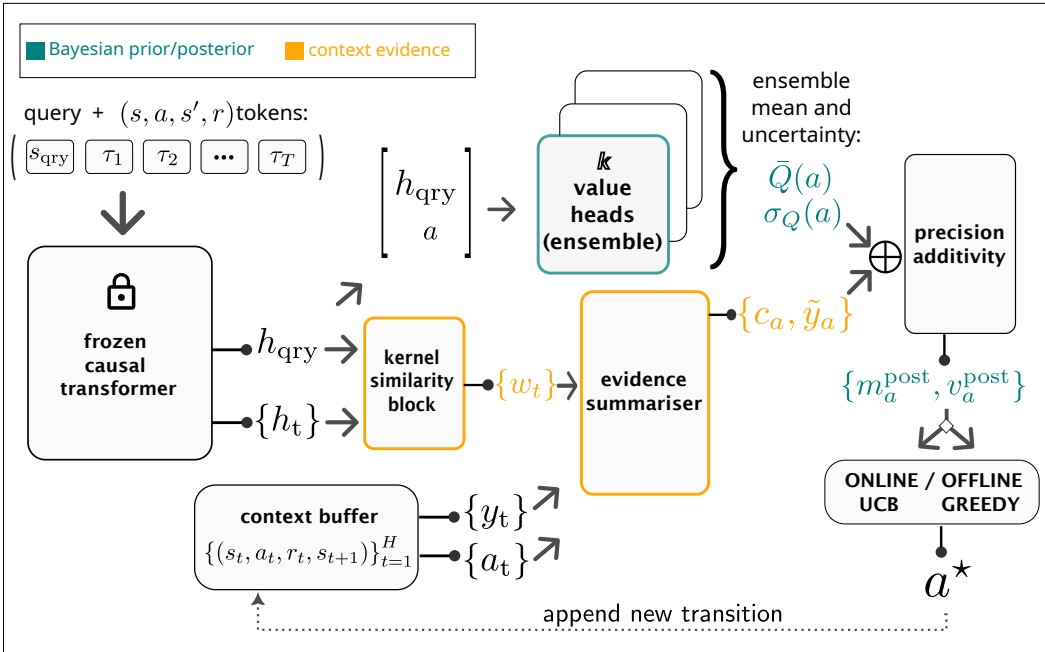

Figure 5: **Detailed test-time architecture diagram** Given a query state $s_{\mathrm{qry}}$ and a multi-episode context buffer of transitions $\{(s_t, a_t, r_t, s_{t+1})\}_{t=1}^T$, a *frozen* causal transformer encodes the query and context into latent features $h_{\mathrm{qry}}$ and $\{h_t\}_{t=1}^T$. Test-time inference decomposes into three stages. **(1) Prior:** a $K$-head value ensemble provides a per-action value prior (ensemble mean and uncertainty), e.g., $(\bar{Q}(a), \sigma_Q^2(a))$. **(2) Evidence:** a kernel similarity module computes weights $\{w_t\}_{t=1}^T$ from latent similarity between $h_{\mathrm{qry}}$ and $\{h_t\}$, and an evidence summariser aggregates the weighted context into action-wise sufficient statistics $(c_a, \tilde{y}_a)$ (pseudo-count and weighted target/return). **(3) Fusion & decision:** a closed-form Bayesian update via precision additivity fuses prior and evidence to produce per-action posterior parameters $(m_a^{\mathrm{post}}, v_a^{\mathrm{post}})$, used for offline greedy selection or online exploration via posterior-UCB to choose $a$. In the online setting, the newest transition is appended to the context buffer and the procedure repeats, enabling gradient-free adaptation driven purely by the evolving context.

**(iii) Epistemic weighting.** Building on ensemble-based uncertainty estimation and randomised priors (Lakshminarayanan et al., 2017; Osband et al., 2018; Pearce et al., 2018; Wilson & Izmailov, 2020), we emphasise samples with higher ensemble standard deviation, concentrating computation on regions of epistemic uncertainty so that the model learns most from poorly covered areas and provides a more informative posterior for exploration:

$$\omega_{\mathrm{epi}} = \mathrm{clip}(1 + \lambda_\sigma\, \sigma_Q(s, a_b^\star),\ \varepsilon,\ c_{\mathrm{epi}}), \tag{19}$$

where $\sigma_Q$ is the ensemble standard deviation from Eq. equation 1. This training-only objective shapes the trunk so that the value ensemble receives features that support calibrated uncertainty and robust value estimation from suboptimal pretraining data.

### A.3 VALUE HEAD TRAINING

**TD(n) targets.** The Q-value ensemble is trained using only the logged context tuples, combining TD($n$) regression Sutton et al. (1998); Dayan (1992) with Bayesian shrinkage Murphy (2007; 2012). For each context window of length $H$ with transitions $\{(s_t, a_t, r_t, s_{t+1})\}_{t=1}^H$, the ensemble mean is

$\bar{Q}(s, a) = \frac{1}{K} \sum_{k=1}^{K} Q_{\phi_k}(s, a)$. A $n$-step bootstrapped targets per time step $t$ can be constructed as:

$$y_t^{(n)} = \sum_{i=0}^{n-1} \gamma^i r_{t+i} + \gamma^n \mathbf{1}[\, t+n \leq H \,] \max_{a'} \bar{Q}(s_{t+n}, a'), \tag{20}$$

where the next $n$ observed rewards are summed along the logged trajectory. A bootstrap term is added only if the context still contains a state $s_{t+n}$[2].

To learn the ensemble, the loss function is composed of two terms $\mathcal{L}_Q = \mathcal{L}_{\text{TD}} + \mathcal{L}_{\text{shrink}}$.

$\mathcal{L}_{\text{TD}}$ - **TD($n$) regression on taken actions.**    For each $(s_t, a_t)$ the ensemble mean is regressed to the TD($n$) target:

$$\mathcal{L}_{\text{TD}} = \mathbb{E}\Big[\big(\bar{Q}(s_t, a_t) - y_t^{(n)}\big)^2\Big]. \tag{21}$$

$\mathcal{L}_{\text{shrink}}$ - **Bayesian shrinkage to per-action posterior means.**    To improve statistical stability, per-action predictions are shrunk toward conjugate posterior means computed from the same TD($n$) targets. For each action $a$ we form counts and empirical TD($n$) averages over the context:

$$c_a = \sum_{t=1}^{H} \mathbb{1}[a_t = a], \qquad \bar{y}_a = \frac{\sum_{t:\, a_t = a} y_t^{(n)}}{\max(1, c_a)}.$$

With prior mean $\mu_0$, prior variance $v_0$, and likelihood variance $\sigma^2$, the per-action posterior mean is

$$m_a^{\text{post}} = \underbrace{\frac{\sigma^2}{\sigma^2 + c_a v_0}}_{w_a} \mu_0 + \big(1 - w_a\big)\, \bar{y}_a, \qquad w_a = \frac{\sigma^2}{\sigma^2 + c_a v_0}. \tag{22}$$

The following loss shrinks the per-action time-average of the ensemble toward $m_a^{\text{post}}$ for actions observed in the context is:

$$\mathcal{L}_{\text{shrink}} = \frac{1}{\sum_a \mathbb{1}[c_a > 0]} \sum_{a:\, c_a > 0} \Big( \underbrace{\frac{1}{H} \sum_{t=1}^{H} \bar{Q}(s_t, a)}_{\text{per-action average over context states}} - m_a^{\text{post}} \Big)^2. \tag{23}$$

**Randomised priors and anchoring.**    Each value head uses a frozen random prior $p_k$ scaled by $\alpha$ and an anchoring penalty that regularises the head's parameters toward their initial values (Osband et al., 2018; Pearce et al., 2018):

$$Q_{\phi_k}(a \mid s, C) = f_k([\mathbf{h}_{\text{qry}};\, \text{onehot}(a)]) + \alpha\, p_k([\mathbf{h}_{\text{qry}};\, \text{onehot}(a)]), \quad \mathcal{L}_{\text{anchor}} = \sum_{k=1}^{K} \sum_j \big\| \phi_{k,j} - \phi_{k,j}^{(0)} \big\|_2^2. \tag{24}$$

**Training Objective**    The full training loss $\mathcal{L} = \mathcal{L}_\pi + \lambda_Q \mathcal{L}_Q + \lambda_{\text{anchor}} \mathcal{L}_{\text{anchor}}$ is optimised using AdamW (Loshchilov & Hutter, 2019). We checkpoint the transformer and heads jointly, and optionally detach policy weights $\omega_b$ during $\mathcal{L}_\pi$ computation to prevent Q-network gradient interference.

## A.4  DERIVATION OF PRECISION-ADDITIVE GAUSSIAN POSTERIOR

This appendix derives the closed-form posterior used in equation 7 in Section 3.3. The result is a standard example of Normal-Normal conjugacy (see Murphy (2007; 2012)) and is included here for completeness and clarity.

**Setup and notation.** At a query state $s$, we aggregate context evidence for each action $a$ using kernel weights $w_t(s) \in [0, 1]$ (defined in Sec. 3.3). We restate the weighted statistics for convenience:

$$c_a(s) = \sum_t w_t(s)\, \mathbb{1}[a_t = a], \qquad \tilde{y}_a(s) = \frac{\sum_t w_t(s)\, \mathbb{1}[a_t = a]\, y_t}{\max(1, c_a(s))},$$

---

[2]Bandits arise as the special case $n=1$ (and $\gamma=0$).

and the target $y_t$ is either the immediate reward or an $n$-step TD target (equation 5).

Under the Gaussian model,

$$Q(a) \sim \mathcal{N}\big(\mu_a^{\mathrm{pri}}, v_a^{\mathrm{pri}}\big), \qquad \tilde{y}_a(s) \mid Q(a) \sim \mathcal{N}\Big(Q(a), \tfrac{\sigma^2}{c_a(s)}\Big),$$

the full weighted least-squares objective $\sum_t w_t(s)\mathbb{1}[a_t{=}a](y_t - Q(a))^2$ decomposes as

$$\sum_t w_t(s)(y_t - Q)^2 = c_a(s)\big(\tilde{y}_a(s) - Q\big)^2 + \sum_t w_t(s)\big(y_t - \tilde{y}_a(s)\big)^2,$$

where the second term is constant in $Q$. Hence $p(\tilde{y}_a(s) \mid Q(a)) \propto \exp\big(-\tfrac{c_a(s)}{2\sigma^2}(Q(a) - \tilde{y}_a(s))^2\big)$. Multiplying by the Gaussian prior and completing the square gives

$$\frac{1}{v_a^{\mathrm{post}}} = \frac{1}{v_a^{\mathrm{pri}}} + \frac{c_a(s)}{\sigma^2}, \qquad m_a^{\mathrm{post}} = v_a^{\mathrm{post}}\left(\frac{\mu_a^{\mathrm{pri}}}{v_a^{\mathrm{pri}}} + \frac{c_a(s)\,\tilde{y}_a(s)}{\sigma^2}\right),$$

which matches Eq. equation 7. See Murphy (2007; 2012) (Normal–Normal conjugacy) for the classical statement.

## A.5 INTUITION AND DESIGN CHOICES

Our goal is to make the model act as if it had a task-specific Bayesian posterior over action values at the query state.

- **Learn a good prior from suboptimal data.** Rather than requiring optimal labels or learning histories, we attach a lightweight ensemble of Q-heads to a DPT-style Transformer trunk. We train this ensemble using TD($n$) regression and Bayesian shrinkage to conjugate per-action means computed from the offline dataset, resulting in a calibrated per-action value prior (mean and variance).

- **Why an ensemble?** Diversity across heads (encouraged by randomised priors and anchoring) captures epistemic uncertainty in areas where the training data provides limited guidance. This uncertainty is needed to perform coherent exploration and for mitigating the effect of suboptimal or incomplete training data.

- **Why a Transformer trunk?** The causal trunk provides a shared representation that conditions on the entire in-task context (state, actions, rewards). This enables the value heads to output prior estimates that are task-aware at the query state, while preserving the simplicity and scalability of sequence modelling.

- **Why train a policy head if we act with the posterior?** We train a policy-head with a propensity-advantage-epistemic weighted cross-entropy loss. Although we do not use this head for control at test time, it corrects the behaviour-policy bias during representation learning, allocated learning capacity to high-value and high-uncertainty examples and co-trains the trunks so that the Q ensemble receives inputs that facilitate reliable value estimation. Decoupling learning (policy supervision improves the trunk) from acting (posterior-UCB uses value uncertainty) is key to achieve robustness from suboptimal training data.

- **Inference time control.** At test time we adapt by performing Bayesian context fusion: we treat the transitions in the context dataset as local evidence about the value of each action near the query state, weight them by similarity to the query (via a kernel) and combine this evidence with the learned value prior. The results is a closed-form posterior mean and variance for every action. This allows the agent to i) exploit the prior knowledge when the context is scarce or empty when interacting with a new environment, ii) update flexibly as more task-specific evidence accumulates and iii) act either conservatively offline (greedy with respect to the posterior mean) or optimistically online (using a UCB rule for exploration). Thus, adaptation produces coherent exploration and strong offline choices entirely through inference, without any gradient updates.

## A.6 ADDITIONAL RELATED WORK

**Uncertainty for exploration in deep RL.** Bootstrapped DQN Osband et al. (2016) and randomised prior functions Osband et al. (2018) introduce randomised value functions and explicit

priors for deep exploration. Deep ensembles provide strong, simple uncertainty estimates Lakshminarayanan et al. (2017), and "anchored" ensembles justify ensembling as approximate Bayesian inference by regularising weights toward prior draws Pearce et al. (2018). SPICE adapts the randomised prior principle to the ICRL setting with an ensemble of value heads and uses a Normal–Normal fusion at test time to produce posterior estimates that feed a UCB-style controller.

Our weighted pretraining objective is conceptually related to advantage-weighted policy learning. AWR performs supervised policy updates with exponentiated advantage weights Peng et al. (2019); AWAC extends this to offline-to-online settings Nair et al. (2020); IQL attains strong offline performance with expectile (upper-value) regression and advantage-weighted cloning Kostrikov et al. (2021). Propensity weighting and counterfactual risk minimisation (IPS/SNIPS/DR) provide a principled basis for importance-weighted objectives under covariate shift Swaminathan & Joachims (2015a;b); Jiang & Li (2016); Thomas & Brunskill (2016). These methods are single-task and do not yield a test-time value posterior for across-task in-context adaptation, which is our focus

**RL via supervised learning and return conditioning.** Beyond DT, the broader RL-via-supervised-learning literature includes return-conditioned supervised learning (RCSL) and analyses of when it recovers optimal policies Brandfonbrener et al. (2022). Implicit Offline RL via Supervised Learning Piche et al. (2022) unifies supervised formulations with implicit models and connects to return-aware objectives. These works motivate our supervised components but do not attach an explicit, calibrated posterior used for a principled controller at test time.

## A.7 PRACTICAL GUIDANCE

- Ensemble size. A small $K$ (e.g., 5–10) already gives reliable uncertainty due to trunk sharing and randomised priors.

- Shrinkage. Moderate shrinkage stabilises training under weak supervision; too much shrinkage can understate uncertainty.

- TD($n$). Larger $n$ reduces bootstrap bias but increases variance; we found mid-range $n$ helpful in sparse-reward MDPs.

- Kernels. Uniform kernels are sufficient for bandits; RBF or cosine kernels help in MDPs with structured state similarity. We primarily use the RBF kernel applied to the latent transformer representation h to leverage the learned, reward-relevant features. The kernel's bandwidth $\tau$ should be tuned to avoid over-smoothing or over-fitting the context evidence.

- Exploration parameter. $\beta_{\text{ucb}}$ tunes optimism; our theory motivates $\beta_t \propto \sqrt{\log t}$, with a fixed $\beta$ working well in short-horizon evaluations.

## A.8 IMPLEMENTATION DETAILS

### A.8.1 BANDIT ALGORITHMS

We follow the baselines and evaluation protocol of Lee et al. (2023). We report offline suboptimality and online cumulative regret, averaging over $N$ tasks; for SPICE and DPT we additionally average over three seeds.

**Empirical Mean (Emp).** Greedy selection by empirical means: $\hat{a} \in \arg\max_a \hat{\mu}_a$, where $\hat{\mu}_a$ is the sample mean of rewards for arm $a$. Offline we restrict to arms observed at least once; online we initialise with one pull per arm (standard good-practice).

**Upper Confidence Bound (UCB).** Optimistic exploration using a Hoeffding bonus. At round $t$, pick $\hat{a} \in \arg\max_a \left( \hat{\mu}_{a,t} + \sqrt{1/n_{a,t}} \right)$, with $n_{a,t}$ pulls of arm $a$. UCB has logarithmic regret in stochastic bandits.

**Lower Confidence Bound (LCB).** Pessimistic selection for offline pick-one evaluation: $\hat{a} \in \arg\max_a \left( \hat{\mu}_a - \sqrt{1/n_a} \right)$. This favours well-sampled actions and is a strong offline baseline when datasets are expert-biased.

**Thompson Sampling (TS)**   Bayesian sampling with Gaussian prior; we set prior mean $1/2$ and variance $1/12$ to match $\mu_a \sim \mathrm{Unif}[0, 1]$ in the DPT setup, and use the correct noise variance at test time.

**DPT.**   Decision-Pretrained Transformer: a GPT-style model trained to predict the optimal action given a query state and an in-context dataset. Offline, DPT acts greedily; online, it samples actions from its policy (as in Lee et al. (2023)), which empirically yields UCB/TS-level exploration and robustness to reward-noise shifts, but only when trained on optimal data.

**SPICE.**   Uncertainty-aware ICRL with a value-ensemble prior and Bayesian test-time fusion. At the query, SPICE forms a per-action posterior from (i) the ensemble prior mean/variance and (ii) state-weighted context statistics, then acts either greedily (offline) or with a posterior-UCB rule (online). The controller attains optimal $O(\log H)$ regret with any prior miscalibration entering only as a constant warm-start term.

### A.8.2   RL Algorithms

We compare to the same meta-RL and sequence-model baselines used in Lee et al. (2023), and deploy SPICE/DPT in the same in-context fashion.

**Proximal Policy Optimisation (PPO).**   Single-task RL trained from scratch (no pretraining); serves as an online-only point of reference for sample efficiency in our few-episode regimes. Hyperparameters follow common practice (SB3 defaults in our code) Schulman et al. (2017).

**Algorithm Distillation (AD).**   A transformer trained via supervised learning on multi-episode learning traces of an RL algorithm; at test time, AD conditions on recent history to act in-context Laskin et al. (2022).

**DPT.**   The same DPT model as described above but applied to MDPs: offline greedy; online sampling from the predicted action distribution each step Lee et al. (2023).

**SPICE.**   The same SPICE controller: posterior-mean (offline) and posterior-UCB (online) built from an ensemble value prior and Bayesian context fusion at test time.

### A.8.3   Bandit Pretraining and Testing

**Task generator and evaluation.**   Each task is a stochastic $A$-armed bandit with $\mu_a \sim \mathrm{Unif}[0, 1]$ and rewards $r \sim \mathcal{N}(\mu_a, \sigma^2)$. Default: $A{=}5$, $H{=}500$, $\sigma{=}0.3$. We report offline suboptimality $\mu^\star - \mu_{\hat{a}}$ vs. context length $h$ and online cumulative regret $\sum_{t=1}^{H}(\mu^\star - \mu_{a_t})$, averaging across $N{=}200$ test environments; for SPICE/DPT we additionally average across 3 seeds and plot $\pm$ SEM bands. For robustness we fix arm means and sweep $\sigma \in \{0.0, 0.3, 0.5\}$.

**Pretraining.**   DPT: 100,000 training bandits; trunk $n_{\text{layer}}{=}6$, $n_{\text{emb}}{=}64$, $n_{\text{head}}{=}1$, dropout $0$, AdamW (lr $= 10^{-4}$), 300 epochs, shuffle, seeds $\{0, 1, 2\}$. **SPICE:** same trunk; $K{=}7$ Q-heads with randomised priors and a small anchor penalty. We optimise a combined objective (policy cross-entropy with propensity/advantage/epistemic weighting for trunk shaping, plus value loss with TD($n$) regression and shrinkage). Unless noted, we use uniform kernel weights for bandits at test time.

**Controllers and deployment.**   Offline: given a fixed context, each method outputs a single arm; SPICE uses $\arg\max_a m_a^{\text{post}}$. Online: methods interact for $H$ steps from empty context; SPICE uses $\arg\max_a \left( m_a^{\text{post}} + \beta \sqrt{v_a^{\text{post}}} \right)$. We match the DPT evaluation by using the same dataset generator, the same number of environments, and identical horizon and noise settings Lee et al. (2023).

**Why SPICE succeeds under weak supervision (intuition).**   The value ensemble provides a calibrated prior that behaves like a small virtual sample count for each arm. Bayesian fusion then combines this prior with weighted empirical evidence, so the posterior rapidly concentrates as data

accrues, shrinking any pretraining bias. Our theory shows this yields $O(\log H)$ **regret with only a constant warm-start penalty** from prior miscalibration; the curves in Fig. 2c–2b mirror this behaviour.

### A.8.4 DARKROOM PRETRAINING AND TESTING

**Environment and data.** We use a continuous darkroom navigation task in which rewards are smooth and peaked around a latent goal location. Each state is represented by a $d$-dimensional feature vector (default $d{=}10$). Actions are discrete with cardinality $A$; dynamics are deterministic given the current state and a one-hot action. For evaluation we generate $N{=}100$ held-out tasks of horizon $H{=}100$ and form an in-context dataset per task consisting of tuples $(s_t, a_t, r_t, s_{t+1})_{t=1}^{H}$. Unless stated otherwise, we use the "weak-last" split from our data generator (the same split is used for all methods). The Darkroom evaluation quantifies distributional shift by holding out 20% of the possible goal locations: 80 unique goals define the training task distribution ($\mathcal{T}_{pre}$), and the remaining 20 unique goals are used for the test task distribution ($\mathcal{T}_{test}$), requiring extrapolation to unseen reward functions. Unless stated otherwise, we use the "weak-last" split from our data generator, meaning that the optimal action label $a^*$ assigned to the query state $s_{qry}$ is simply the last action ($a_H$) that occurred in the in-context trajectory, $C$. This action is typically suboptimal and provides an explicitly suboptimal supervision.

**Pretraining.** Both SPICE and DPT share the same GPT-style trunk ($n_{\text{layer}}{=}6$, $n_{\text{emb}}{=}64$, $n_{\text{head}}{=}1$, dropout 0), trained with AdamW at learning rate $10^{-4}$ for 50 epochs.[3] DPT is trained with the standard DPT objective on 100,000 darkroom tasks (shuffled mini-batches). SPICE attaches an ensemble of $K{=}7$ value heads with randomised priors and trains them via TD($n$) regression with $n{=}5$ and $\gamma{=}0.95$, plus conjugate shrinkage and a small anchor penalty (see Alg. 1). All hyperparameters used by the test-time Bayesian fusion are fixed a priori: RBF kernel with scale $\tau{=}0.5$, evidence noise $\sigma^2{=}0.09$, and prior-variance floor $v_{\min}{=}10^{-2}$.

**Controllers at test time.** For SPICE we evaluate posterior-UCB with three optimism levels, $\beta \in \{0.5, 1.0, 2.0\}$; the offline analogue uses the posterior mean (greedy). For DPT we use the greedy controller that selects $\arg\max_a$ of the policy logits at the query state. When averaging across seeds, we first average per task across the three checkpoints and then aggregate across tasks; error bands report $\pm$ SEM.

**Evaluation protocol.** We report two metrics: (i) *Online return*: (ii) *Online cumulative regret*: starting from an empty context, a controller interacts for $H$ steps; at each step we compare the reward of the chosen action to the reward of the environment's optimal action at the same state. To ensure a fair comparison, for each held-out task we draw a single initial state $s_0$ and use it for all controllers and seeds before averaging. For each metric we average across the $N{=}100$ held-out tasks. We average over three seeds. Shaded regions denote $\pm$ SEM across tasks.

## B PROOF OF THEOREM 1

**Proof Overview.** We analyse the posterior-UCB controller by (i) treating the ensemble prior at the query as a Normal prior with mean $\mu_a^{\text{pri}}$ and variance $v_a^{\text{pri}}$, resulting in a posterior with pseudo-count $N_a^{\text{pri}} = \sigma^2/v_a^{\text{pri}}$ under Normal-Normal conjugacy (Murphy, 2007; 2012); (ii) showing that the posterior mean is a convex combination of the empirical and prior means and the posterior variance shrinks at least as $O(1/n_{a,t})$ (Lemma 1); and (iii) combining sub-Gaussian concentration (Hoeffding-style) with a UCB schedule $\beta_t = \sqrt{2 \log t}$ (Hoeffding, 1963; Auer et al., 2002) to upper-bound pulls of suboptimal arms. This results in $O(\log K)$ regret plus a constant warm-start term proportional to $N_a^{\text{pri}}|\mu_a^{\text{pri}} - \mu_a|$ (Lemma 2), recovering classical UCB when the prior is uninformative or well calibrated.

---

[3]We train three seeds for each method; checkpoints are averaged only at evaluation time.

For completeness, we re-state the theorem here: *Under the assumption of $\sigma^2$-sub-Gaussian reward distributions, the SPICE inference controller satisfies*

$$\mathbb{E}\Big[ \sum_{t=1}^{K}(\mu_\star - \mu_{a_t}) \Big] \leq \sum_{a \neq \star} \Big( \frac{32\sigma^2 \log K}{\Delta_a} + 4N_a^{\text{pri}}\big|\mu_a^{\text{pri}} - \mu_a\big| \Big) + O(1).$$

First, we consider the following lemmas

**Lemma 1** (Bias-variance decomposition). *With the posterior defined in Eq. 11, for all $a, t$ it holds that*

$$\Big|m_{a,t}^{post} - \mu_a\Big| \leq \Big|\widehat{\mu}_{a,t} - \mu_a\Big| + \frac{N_a^{pri}}{N_a^{pri} + n_{a,t}}\Big|\mu_a^{pri} - \mu_a\Big|, \quad v_{a,t}^{post} \leq \frac{\sigma^2}{n_{a,t}} \tag{25}$$

This lemma shows that the posterior mean forms a weighted average of the empirical and prior means, with relative error decomposing into two components: a variance term $\big|\widehat{\mu}_{a,t} - \mu_a\big|$ capturing finite-sample noise in the empirical mean, and a bias term $\frac{N_a^{\text{pri}}}{N_a^{\text{pri}}+n_{a,t}}\big|\mu_a^{\text{pri}} - \mu_a\big|$ reflecting prior miscalibration. As $n_{a,t} \to \infty$, the bias term vanishes, eliminating prior miscalibration, while the posterior variance shrinks at least as fast as the frequentist variance $\frac{\sigma^2}{n_{a,t}}$ (see Eq. 25).

*Proof.* The posterior mean for arm $a$ at round $t$ from equation Eq. 11 can be rewritten as a convex combination

$$m_{a,t}^{\text{post}} = \alpha_{a,t}\widehat{\mu}_{a,t} + (1 - \alpha_{a,t})\mu_a^{\text{pri}}, \quad \alpha_{a,t} = \frac{n_{a,t}}{N_a^{\text{pri}} + n_{a,t}}$$

Subtracting the true mean $\mu_a$ gives

$$m_{a,t}^{\text{post}} - \mu_a = \alpha_{a,t}(\widehat{\mu}_{a,t} - \mu_a) + (1 - \alpha_{a,t})(\mu_a^{\text{pri}} - \mu_a)$$

Taking absolute values and applying the triangle inequality gives

$$\Big|m_{a,t}^{\text{post}} - \mu_a\Big| \leq \alpha_{a,t}\Big|(\widehat{\mu}_{a,t} - \mu_a)\Big| + (1 - \alpha_{a,t})\Big|(\mu_a^{\text{pri}} - \mu_a)\Big|$$

Since $\alpha_{a,t} \leq 1$ we can drop the factor and since $1 - \alpha_{a,t} = \frac{N_a^{\text{pri}}}{N_a^{\text{pri}}+n_{a,t}}$, we obtain

$$\Big|m_{a,t}^{\text{post}} - \mu_a\Big| \leq \Big|(\widehat{\mu}_{a,t} - \mu_a)\Big| + \frac{N_a^{\text{pri}}}{N_a^{\text{pri}} + n_{a,t}}\Big|(\mu_a^{\text{pri}} - \mu_a)\Big|$$

Since $N_a^{\text{pri}} \geq 0$ we get

$$v_{a,t}^{\text{post}} = \frac{\sigma^2}{N_a^{\text{pri}} + n_{a,t}} \leq \frac{\sigma^2}{n_{a,t}}.$$

$\square$

**Lemma 2** (Posterior concentration). *We fix a horizon $K \geq 2$. Under the assumption of $\sigma^2$-sub-Gaussian reward distributions, the following inequality holds simultaneously for all arms $a$ and all rounds $t \in \{1, ..., K\}$ with probability at least $1 - O(\frac{1}{K})$*

$$\mu_a \leq m_{a,t}^{post} + \beta_t\sqrt{v_{a,t}^{post}} + \frac{N_a^{pri}}{N_a^{pri} + n_{a,t}}\Big|\mu^{pri} - \mu_a\Big|, \quad \beta_t = \sqrt{2\log t}$$

*Note that this also yields a symmetric lower bound with the last two terms negated.*

*Proof.* Using Eq. 10 and a union bound over all $a$ and $t \leq K$ we obtain the following bound with probability at least $1 - O(1/K)$

$$|\widehat{\mu}_{a,t} - \mu_a| \leq \sigma\sqrt{\frac{2\log t}{n_{a,t}}} \quad \text{for all } a \text{ and } t \leq K$$

Since $v_{a,t}^{\text{post}} = \frac{\sigma^2}{N_a^{\text{pri}} + n_{a,t}} \leq \frac{\sigma^2}{n_{a,t}}$ we get

$$\sigma\sqrt{\frac{2\log t}{n_{a,t}}} \leq \sqrt{2\log t}\sqrt{v_{a,t}^{\text{post}}} = \beta_t\sqrt{v_{a,t}^{\text{post}}}$$

and thus

$$|\widehat{\mu}_{a,t} - \mu_a| \leq \beta_t\sqrt{v_{a,t}^{\text{post}}}.$$

Combining that with Lemma 1 gives

$$\left|m_{a,t}^{\text{post}} - \mu_a\right| \leq \beta_t\sqrt{v_{a,t}^{\text{post}}} + \frac{N_a^{\text{pri}}}{N_a^{\text{pri}} + n_{a,t}}\left|\mu_a^{\text{pri}} - \mu_a\right|.$$

Expanding this absolute value bound into one-sided inequalities yields the result. $\qquad\square$

Using these lemmas, the proof of Theorem 1 follows.

*Proof.* Let $N_a(K) := \sum_{t=1}^{K}\mathbf{1}\{a_t = a\}$ be the pull count of arm $a$ up to horizon $K$. We can decompose the regret as $\mathbb{E}\left[\sum_{t=1}^{K}(\mu_\star - \mu_{a,t})\right] = \sum_{a\neq\star}\Delta_a\mathbb{E}\left[N_a(K)\right]$. We derive an upper bound for $N_a(K)$ for each suboptimal arm $a$.

Consider an horizon $K \geq 2$, define the good event for each arm $a \in [A]$ and step $t \in \{1, ..., K\}$

$$G_{a,t} := \left\{\left|\widehat{\mu}_{a,t} - \mu_a\right| \leq \sigma\sqrt{\frac{2\log t}{n_{a,t}}}\right\}$$

$G_{a,t}$ is the event that the empirical mean of arm $a$ at time $t$ lies within its confidence interval. We define the event that concentration holds for all arms and times simultaneously

$$\mathcal{E} := \bigcap_{a=1}^{A}\bigcap_{t=1}^{K} G_{a,t}.$$

The complement corresponds to the event that concentration fails for at least one $(a,t)$

$$\mathcal{E}^c := \bigcup_{a=1}^{A}\bigcup_{t=1}^{K} G_{a,t}^c.$$

Using the union bound, we get

$$\Pr(\mathcal{E}^c) \leq \sum_{a=1}^{A}\sum_{t=1}^{K}\Pr(G_{a,t}^c) \leq \sum_{a=1}^{A}\sum_{t=1}^{K}\frac{2}{t^2} \leq 2A\sum_{t=1}^{\infty}\frac{1}{t^2} = \frac{\pi^2}{3}A.$$

If we instead define $G_{a,t}$ using an inflated radius $\sigma\sqrt{\frac{2\log(cAK^2)}{n_{a,t}}}$ we similarly get $\Pr(\mathcal{E}^c) \leq O(\frac{1}{K})$.

We decompose the regret as

$$\mathbb{E}[R_K] = \mathbb{E}[R_K \mid \mathcal{E}]\Pr(\mathcal{E}) + \mathbb{E}[R_K \mid \mathcal{E}^c]\Pr(\mathcal{E}^c) \leq \mathbb{E}[R_K \mid \mathcal{E}] + K\Pr(\mathcal{E}^c) \leq \mathbb{E}[R_K \mid \mathcal{E}] + O(1),$$

so it is sufficient to bound the regret on $\mathcal{E}$.

Using Lemma 1, knowing that $\sigma\sqrt{2\log t/n_{a,t}} \leq \beta_t\sqrt{v_{a,t}^{\text{post}}}$ and that $|\widehat{\mu}_{a,t} - \mu_a| \leq \sigma\sqrt{2\log t/n_{a,t}}$ for $\mathcal{E}$, we get

$$\mu_a \leq m_{a,t}^{\text{post}} + \beta_t\sqrt{v_{a,t}^{\text{post}}} + \frac{N_a^{\text{pri}}}{N_a^{\text{pri}} + n_{a,t}}|\mu_a^{\text{pri}} - \mu_a|, \quad \beta_t = \sqrt{2\log t}. \tag{26}$$

Suppose we pick a suboptimal arm $a \neq \star$ at round $t$. Using Eq. 26 for $a$ and $\star$ as well as the SPICE selection rule $m_{a,t-1}^{\text{post}} + \beta_t \sqrt{v_{a,t-1}^{\text{post}}} \geq m_{\star,t-1}^{\text{post}} + \beta_t \sqrt{v_{\star,t-1}^{\text{post}}}$ we get

$$\Delta_a \leq 2\beta_t \sqrt{v_{a,t-1}^{\text{post}}} + \frac{N_a^{\text{pri}}}{N_a^{\text{pri}} + n_{a,t}}|\mu_a^{\text{pri}} - \mu_a| + \frac{N_\star^{\text{pri}}}{N_\star^{\text{pri}} + n_{\star,t}}|\mu_\star^{\text{pri}} - \mu_\star|. \tag{27}$$

First, we derive a threshold for the variance term in Eq. 27. Using $\beta_t = \sqrt{2\log t}$, Lemma 1 and Eq. 25) we obtain

$$2\beta_t \sqrt{v_{a,t-1}^{\text{post}}} \leq 2\sqrt{2\log t}\frac{\sigma}{\sqrt{n_{a,t-1}}}.$$

Using a similar technique as in the classical UCB1 proof Auer et al. (2002) we make the variance term smaller than half the gap $\Delta_a/2$

$$2\sqrt{2\log t}\frac{\sigma}{\sqrt{n_{a,t-1}}} \leq \frac{\Delta_a}{2} \quad \Rightarrow \quad n_{a,t-1} \geq \frac{32\sigma^2 \log t}{\Delta_a^2}$$

As $t \leq K$ we can replace $\log t$ with the worst-case $logK$ to ensure that the condition holds for all rounds up to horizon $K$. The variance threshold is therefore

$$n_a^\dagger := \left\lceil \frac{32\sigma^2 \log K}{\Delta_a^2} \right\rceil.$$

Once arm $a$ has been pulled at least $n_a^\dagger$ times, the variance term $2\beta_t \sqrt{v_{a,t-1}^{\text{post}}}$ in Eq. 27 is guaranteed to be at most $\Delta_a/2$ for every $t \leq K$.

Second, we derive a threshold for the prior bias terms. To force the prior bias term below $\Delta_a/4$ we define $\delta_a := |\mu_a^{\text{pri}} - \mu_a|$ and solve

$$\frac{N_a^{\text{pri}}}{N_a^{\text{pri}} + n_{a,t}}\delta_a \leq \frac{\Delta_a}{4} \quad \Rightarrow \quad n_{a,t-1} \geq \frac{4N_a^{\text{pri}}\delta_a}{\Delta_a} - N_a^{\text{pri}} \leq \frac{4N_a^{\text{pri}}\delta_a}{\Delta_a}.$$

Thus after about

$$n_a^{pri} := \left\lceil \frac{4N_a^{\text{pri}}\delta_a}{\Delta_a} \right\rceil$$

pulls of arm $a$ its prior bias term is guaranteed to be below $\Delta_a/4$. The same argument applies to the optimal arm $\star$: its prior bias terms decreases as $n_{\star,t}$ grows and since $\star$ is selected frequently, only a constant number of pulls ins needed before its prior bias term is below $\Delta_A/4$.

By combining the bias and variance thresholds, we can derive the following bound for $N_a(K)$ under the event $\mathcal{E}$ for some constant $C_a$ (independent of $K$)

$$\mathbb{E}[N_a(K)\mathbf{1}_{\mathcal{E}}] \leq n_a^\dagger + n_a^{\text{pri}} + C_a \leq \frac{32\sigma^2 \log K}{\Delta_a^2} + \frac{4N_a^{\text{pri}}\delta_a}{\Delta_a} + C_a.$$

By multiplying by $\Delta_a$ and summing over all arms $a \neq \star$ we obtain

$$\mathbb{E}[R_K\mathbf{1}_{\mathcal{E}}] \leq \sum_{a\neq\star} \frac{32\sigma^2 \log K}{\Delta_a^2} + \sum_{a\neq\star} 4N_a^{\text{pri}}\delta_a + O(1).$$

To conclude, we collect all bounded terms and include the contribution of the event $\mathcal{E}^c$ into an $O(1)$ term to obtain

$$\mathbb{E}\Big[\sum_{t=1}^{K}(\mu_\star - \mu_{a,t})\Big] \leq \sum_{a\neq\star} \frac{32\sigma^2 \log K}{\Delta_a^2} + \sum_{a\neq\star} 4N_a^{\text{pri}}|\mu_a^{pri} - \mu_a| + O(1).$$

$\square$

## C   PROOF OF THEOREM 2

**Proof Overview.**   We extend the bandit analysis of Theorem 1 to the MDP setting by (i) treating the sequence of kernel-weighted TD targets as a noisy observation of $Q_\star(s,a)$ for each state-action pair (s,a) and showing that the SPICE posterior matches the bandit posterior of Definition 1; (ii) obtaining bandit-style high-probability confidence intervals for every (s,a), resulting in an upper confidence bound of the form $Q_\star(s,a) \leq m^{post}_{(s,a,t)} + \beta_t \sqrt{v^{post}_{(s,a,t)}} +$ (prior bias); (iii) using these confidence intervals in a standard UCB-style regret decomposition for finite-horizon MDPs, which bounds per-episode regret by the sum of confidence bonuses along the visited trajectory; (iv) summing the UCB bonuses over all episodes to obtain the $O(H\sqrt{SAK})$ term; and (v) showing that prior miscalibration only contributes an additive $O(N^{pri}_{(s,a)}|\mu^{pri}_{s,a} - Q_\star(s,a)|)$ term per state-action pair.

For completeness, we re-state the theorem here:

[SPICE's Regret-optimality in Finite-Horizon MDPs] *Consider a finite-horizon MDP $M = \langle \mathcal{S}, \mathcal{A}, T, R, H \rangle$ with finite state and action spaces $|\mathcal{S}| = S$, $|\mathcal{A}| = A$, bounded rewards $r \in [0,1]$ and fixed episode length $H$. We write $K$ for the number of episodes and $T := KH$ for the total number of interaction steps. Assume that for every $(s,a)$ there exists an $n$ such that the kernel-weighted average of the $n$-step TD targets $y^{(n)}_t$ (Definition 3) satisfies*

$$\mathbb{E}\big[y^{(n)}_t \mid s_t = s, a_t = a, \mathcal{F}_{t-1}\big] = Q_\star(s,a), \qquad y^{(n)}_t - Q_\star(s,a) \text{ is conditionally } \sigma_Q\text{-sub-Gaussian},$$

*for some variance proxy $\sigma_Q^2 \leq c_H H$ depending only on the horizon, where $\mathcal{F}_{t-1}$ is the history up to time $t-1$. Let the SPICE inference controller maintain for each $(s,a)$ a Gaussian prior $Q(s,a) \sim \mathcal{N}(\mu^{pri}_{s,a}, v^{pri}_{s,a})$ and act with the posterior-UCB rule*

$$a_t \in \arg\max_{a \in \mathcal{A}} \big\{ m^{post}_{s_t,a,t} + \beta_t \sqrt{v^{post}_{s_t,a,t}} \big\}.$$

*Let $N^{pri}_{s,a} := \sigma_Q^2 / v^{pri}_{s,a}$ be the prior pseudo-count and denote $N^{max} := \max_{s,a} N^{pri}_{s,a}$. We assume an exploration schedule of the form*

$$\beta_t := C_\beta \sqrt{\log(SAT)}, \qquad C_\beta \geq 2\sqrt{1 + N^{max}},$$

*which is of order $\Theta(\sqrt{\log T})$ and whose constant depends only on the prior. Then the cumulative regret over $K$ episodes satisfies*

$$\mathbb{E}[\text{Regret}_K] = \mathbb{E}\Big[\sum_{k=1}^{K} \big(V_\star(s_1^k) - V_{\pi_k}(s_1^k)\big)\Big] \leq O\big(H\sqrt{SAK}\big) + \sum_{(s,a) \in \mathcal{S} \times \mathcal{A}} O\Big(N^{pri}_{s,a}|\mu^{pri}_{s,a} - Q_\star(s,a)|\Big), \tag{28}$$

*where $\pi_k$ is the policy used in episode $k$.*

**Notation.**   We consider $K$ episodes, each of horizon $H$ and $T = KH$. We index time by the pair $(k,h)$ where $k \in \{1, ..., K\}$ is the episode and $h \in \{1, ..., H\}$ is the within-episode step. Let $\pi_k$ be the policy used in episode $k$ by the SPICE controller. For each $(s,a)$ we write $Q_\star(s,a)$ for the optimal Q-value and $V_\star(s,a) = \max_a Q_\star(s,a)$ and $V_{\pi_k}(s)$ for the value of policy $\pi_k$ from state $s$. Let $N_{s,a,t}$ be the number of times the pair $(s,a)$ has been visited up to (and including) global time $t$. Whenever $(s,a)$ is executed, SPICE constructs an n-step TD target $y^{(n)}_t$ as in Definition 3. We assume the kernel-weighted average of these targets gives a $\sigma_Q^2$-sub-Gaussian estimator of $Q_\star(s,a)$:

$$\mathbb{E}\big[y^{(n)}_t \mid s_t = s, a_t = a, \mathcal{F}_{t-1}\big] = Q_\star(s,a),$$
$$y^{(n)}_t - Q_\star(s,a) \text{ is } \sigma_Q\text{-sub-Gaussian with variance proxy } \sigma_Q^2 \leq c_H H. \tag{29}$$

For each pair $(s,a)$ SPICE maintains a Gaussian prior $Q(s,a) \sim \mathcal{N}(\mu^{pri}_{s,a}, v^{pri}_{s,a})$ and updates this prior using the kernel-weighted TD targets and a Gaussian likelihood with variance $\sigma_Q^2$ (cf Equation 7). By Normal-Normal conjugacy we obtain the same formula as in Definition 1, now indexed

by $(s, a)$:

$$N_{s,a}^{\text{pri}} := \frac{\sigma_Q^2}{v_{s,a}^{\text{pri}}}, \qquad m_{s,a,t}^{\text{post}} = \frac{N_{s,a}^{\text{pri}}\mu_{s,a}^{\text{pri}} + N_{s,a,t}\,\bar{y}_{s,a,t}}{N_{s,a}^{\text{pri}} + N_{s,a,t}}, \qquad v_{s,a,t}^{\text{post}} = \frac{\sigma_Q^2}{N_{s,a}^{\text{pri}} + N_{s,a,t}}, \qquad (30)$$

where $\bar{y}_{s,a,t}$ is the empirical average of the TD targets obtained for $(s, a)$ up to time $t$.

First, we consider the following lemmas.

**Lemma 3** (MDP posterior-variance decomposition.). *With the posterior defined in equation 30, for all $(s, a)$ and all times $t \geq 1$ it holds that*

$$\left| m_{s,a,t}^{\text{post}} - Q_\star(s,a) \right| \leq \left| \bar{y}_{s,a,t} - Q_\star(s,a) \right| + \frac{N_{s,a}^{\text{pri}}}{N_{s,a}^{\text{pri}} + N_{s,a,t}} \left| \mu_{s,a}^{\text{pri}} - Q_\star(s,a) \right|, \qquad v_{s,a,t}^{\text{post}} \leq \frac{\sigma_Q^2}{N_{s,a,t}}.$$
$$(31)$$

*Proof.* The posterior in equation 30 matches the SPICE posterior for a bandit arm with prior mean $\mu_{s,a}^{pri}$, prior variance $v_{s,a}^{pri}$, sub-Gaussian noise level $\sigma_Q^2$ and sample mean $\bar{y}_{s,a,t}$. Applying Lemma 1 with $\mu_a \leftarrow Q_\star(s,a)$, $\hat{\mu}_{a,t} \leftarrow \bar{y}_{s,a,t}$ and $n_{a,t} \leftarrow N_{s,a,t}$ gives exactly the inequality in Lemma 3. $\square$

**Lemma 4** (MDP posterior concentration.). *We fix a horizon $K \geq 2$ and set $T = KH$. Under the assumptions stated in equation 29 and with the exploration schedule $\beta_t$ defined in Theorem 2, it holds with probability at least $1 - O(1/K)$, simultaneously for all $(s, a) \in \mathcal{S} \times \mathcal{A}$ and all $t \leq T$ that*

$$Q_\star(s,a) \leq m_{s,a,t}^{\text{post}} + \beta_t \sqrt{v_{s,a,t}^{\text{post}}} + \frac{N_{s,a}^{\text{pri}}}{N_{s,a}^{\text{pri}} + N_{s,a,t}} \left| \mu_{s,a}^{\text{pri}} - Q_\star(s,a) \right|, \qquad (32)$$

*and an analogous lower bound holds with the last two terms negated.*

*Proof.* We fix $(s, a)$ and $t$. Conditioned on the event $\{N_{s,a,t} = n\}$, the empirical average $\bar{y}_{s,a,t}$ is the mean of $n$ independent conditionally $\sigma_Q$-sub-Gaussian variables with mean $Q_\star(s,a)$. By a standard sub-Gaussian tail bound Rebeschini (2021) we have for all $\epsilon > 0$,

$$\mathbb{P}\left( \left| \bar{y}_{s,a,t} - Q_\star(s,a) \right| > \epsilon \,\big|\, N_{s,a,t} = n \right) \leq 2\exp\left( -\frac{n\epsilon^2}{2\sigma_Q^2} \right).$$

We set

$$\epsilon_n := \sigma_Q \sqrt{\frac{4\log(SAT)}{n}}.$$

Then

$$\mathbb{P}\left( \left| \bar{y}_{s,a,t} - Q_\star(s,a) \right| > \sigma_Q \sqrt{\tfrac{4\log(SAT)}{N_{s,a,t}}} \right) \leq \frac{2}{(SAT)^2},$$

Applying the tail bound to each fixed $(s, a, t)$ we obtain

$$\mathbb{P}(E_{s,a,t}) \leq \frac{2}{(SAT)^2}, \qquad E_{s,a,t} := \left\{ \left| \bar{y}_{s,a,t} - Q_\star(s,a) \right| > \sigma_Q \sqrt{\tfrac{4\log(SAT)}{N_{s,a,t}}} \right\}.$$

Using the union bound over all $SAT$ triples $(s, a, t)$ gives

$$\mathbb{P}\left( \bigcup_{s,a,t} E_{s,a,t} \right) \leq \sum_{s,a,t} \mathbb{P}(E_{s,a,t}) \leq (SAT) \cdot \frac{2}{(SAT)^2} = \frac{2}{SAT}.$$

Equivalently,

$$\mathbb{P}\left( \bigcap_{s,a,t} E_{s,a,t}^c \right) = 1 - \mathbb{P}\left( \bigcup_{s,a,t} E_{s,a,t} \right) \geq 1 - \frac{2}{SAT}.$$

Since $T = KH$, we have

$$\frac{2}{SAT} = \frac{2}{SA \cdot KH} = O\left(\frac{1}{K}\right),$$

and therefore, with probability at least $1 - O(1/K)$,

$$\left|\bar{y}_{s,a,t} - Q_\star(s,a)\right| \leq \sigma_Q\sqrt{\frac{4\log(SAT)}{N_{s,a,t}}} \quad \text{for all } (s,a,t) \text{ simultaneously.}$$

Next we relate this empirical radius to the posterior variance. From equation 30 we have

$$v_{s,a,t}^{\text{post}} = \frac{\sigma_Q^2}{N_{s,a}^{\text{pri}} + N_{s,a,t}}, \qquad \sqrt{v_{s,a,t}^{\text{post}}} = \frac{\sigma_Q}{\sqrt{N_{s,a}^{\text{pri}} + N_{s,a,t}}}.$$

Therefore,

$$\sigma_Q\sqrt{\frac{4\log(SAT)}{N_{s,a,t}}} = \sqrt{4\log(SAT)}\,\frac{\sigma_Q}{\sqrt{N_{s,a}^{\text{pri}} + N_{s,a,t}}}\sqrt{1 + \frac{N_{s,a}^{\text{pri}}}{N_{s,a,t}}} = \sqrt{4\log(SAT)}\,\sqrt{v_{s,a,t}^{\text{post}}}\,\sqrt{1 + \frac{N_{s,a}^{\text{pri}}}{N_{s,a,t}}}.$$

For any $n \geq 1$ we have

$$\sqrt{1 + N_{s,a}^{\text{pri}}/n} \leq \sqrt{1 + N^{\max}}.$$

By construction we choose $C_\beta \geq 2\sqrt{1 + N^{\max}}$, so

$$\sqrt{4\log(SAT)}\sqrt{1 + \frac{N_{s,a}^{\text{pri}}}{N_{s,a,t}}} \leq C_\beta\sqrt{\log(SAT)} = \beta_t.$$

Therefore,

$$\left|\bar{y}_{s,a,t} - Q_\star(s,a)\right| \leq \beta_t\sqrt{v_{s,a,t}^{\text{post}}}$$

for all $(s,a,t)$ with the probability at least $1 - O(1/K)$. Combining this with Lemma 3 gives

$$\left|m_{s,a,t}^{\text{post}} - Q_\star(s,a)\right| \leq \beta_t\sqrt{v_{s,a,t}^{\text{post}}} + \frac{N_{s,a}^{\text{pri}}}{N_{s,a}^{\text{pri}} + N_{s,a,t}}\left|\mu_{s,a}^{\text{pri}} - Q_\star(s,a)\right|,$$

which implies the inequality 32 (and its lower-tail analogue) by expanding the absolute value. $\qquad\square$

**Notation.** We introduce the shorthand

$$\text{bias}_{s,a,t} := \frac{N_{s,a}^{\text{pri}}}{N_{s,a}^{\text{pri}} + N_{s,a,t}}\left|\mu_{s,a}^{\text{pri}} - Q_\star(s,a)\right|.$$

From Lemma 4 we thus have, for all $(s,a,t)$,

$$Q_\star(s,a) \leq m_{s,a,t}^{\text{post}} + \beta_t\sqrt{v_{s,a,t}^{\text{post}}} + \text{bias}_{s,a,t}. \tag{33}$$

with probability at least $1 - O(1/K)$.

**Definition 4** (Optimistic Q values.). *For each global time $t$ and state-action pair $(s,a)$ we write*

$$\widetilde{Q}_t(s,a) := m_{s,a,t}^{\text{post}} + \beta_t\sqrt{v_{s,a,t}^{\text{post}}}, \qquad \widetilde{V}_t(s) := \max_{a\in\mathcal{A}}\widetilde{Q}_t(s,a).$$

*The SPICE controller at time $t$ in state $s_t$ chooses*

$$a_t \in \arg\max_{a\in\mathcal{A}}\widetilde{Q}_t(s_t,a),$$

*i.e. it is greedy with respect to $\widetilde{Q}_t$.*

On the high-probability event of Lemma 4, the optimistic Q-values upper bound the optimal Q-values, up to an additional term caused by prior miscalibration.

**Definition 5** (Optimism of SPICE Q-values.). *On the high-probability event of Lemma 4, for all $(s, a, t)$,*

$$Q_\star(s, a) \leq \widetilde{Q}_t(s, a) + \text{bias}_{s,a,t}. \tag{34}$$

*Therefore,*

$$V_\star(s) = \max_a Q_\star(s, a) \leq \widetilde{V}_t(s) + \max_a \text{bias}_{s,a,t}. \tag{35}$$

*Proof.* The inequality 34 is a direct rewriting of equation 33 with $\widetilde{Q}_t(s, a)$ substituted. The bound on $V_\star$ follows by taking maxima over $a$. $\square$

**Lemma 5** (Per-episode regret decomposition.). *On the event of Lemma 4, the regret in episode $k$ satisfies*

$$V_\star(s_1^k) - V_{\pi_k}(s_1^k) \leq C_0 \sum_{h=1}^{H} \left( \beta_{t_{k,h}} \sqrt{v_{s_{k,h}, a_{k,h}, t_{k,h}}^{\text{post}}} + \text{bias}_{s_{k,h}, a_{k,h}, t_{k,h}} \right), \tag{36}$$

*for some universal constant $C_0 > 0$, where $t_{k,h}$ denotes the global time index corresponding to step $(k, h)$.*

*Proof.* Let

$$\Delta_{k,h}(s) := V_{\star,h}(s) - V_{\pi_k, h}(s)$$

denote the difference between the optimal $h$-step value and the value of policy $\pi_k$ from state $s$ with $h$ steps remaining. We write $V_{\star, H+1} = V_{\pi_k, H+1} \equiv 0$ since at step $H + 1$ the episode has terminated and no future rewards can be collected. We fix episode $k$ and consider step $h$ with state $s_{k,h}$ and the action $a_{k,h}$ chosen by SPICE. By definition,

$$V_{\pi_k, h}(s_{k,h}) = Q_{\pi_k, h}(s_{k,h}, a_{k,h}).$$

For the chosen pair $(s_{k,h}, a_{k,h})$ we have, by Lemma 4,

$$\left| \widetilde{Q}_{t_{k,h}}(s_{k,h}, a_{k,h}) - Q_{\star,h}(s_{k,h}, a_{k,h}) \right| \leq \beta_{t_{k,h}} \sqrt{v_{s_{k,h}, a_{k,h}, t_{k,h}}^{\text{post}}} + \text{bias}_{s_{k,h}, a_{k,h}, t_{k,h}}.$$

Furthermore, by the Bellman equations for the optimal value function and for the value of policy $\pi_k$, we have

$$Q_{\star,h}(s_{k,h}, a_{k,h}) = \mathbb{E}\left[ r_{k,h} + V_{\star,h+1}(s_{k,h+1}) \mid s_{k,h}, a_{k,h} \right],$$

$$Q_{\pi_k, h}(s_{k,h}, a_{k,h}) = \mathbb{E}\left[ r_{k,h} + V_{\pi_k, h+1}(s_{k,h+1}) \mid s_{k,h}, a_{k,h} \right].$$

Subtracting the second identity from the first and using linearity of conditional expectation yields

$$Q_{\star,h}(s_{k,h}, a_{k,h}) - Q_{\pi_k, h}(s_{k,h}, a_{k,h}) = \mathbb{E}\left[ r_{k,h} + V_{\star,h+1}(s_{k,h+1}) - \left( r_{k,h} + V_{\pi_k, h+1}(s_{k,h+1}) \right) \mid s_{k,h}, a_{k,h} \right]$$

$$= \mathbb{E}\left[ V_{\star,h+1}(s_{k,h+1}) - V_{\pi_k, h+1}(s_{k,h+1}) \mid s_{k,h}, a_{k,h} \right].$$

Recalling the definition

$$\Delta_{k,h+1}(s) := V_{\star,h+1}(s) - V_{\pi_k, h+1}(s),$$

we can rewrite this as

$$Q_{\star,h}(s_{k,h}, a_{k,h}) - Q_{\pi_k, h}(s_{k,h}, a_{k,h}) = \mathbb{E}\left[ \Delta_{k,h+1}(s_{k,h+1}) \mid s_{k,h}, a_{k,h} \right].$$

We now decompose the suboptimality at step $h$ in episode $k$. At the visited state $s_{k,h}$ we have

$$\Delta_{k,h}(s_{k,h}) = V_{\star,h}(s_{k,h}) - V_{\pi_k, h}(s_{k,h}).$$

At step $h$, the policy $\pi_k$ chooses action $a_{k,h}$ in state $s_{k,h}$, hence

$$V_{\pi_k, h}(s_{k,h}) = Q_{\pi_k, h}(s_{k,h}, a_{k,h}).$$

Using this and adding and subtracting $Q_{\star,h}(s_{k,h}, a_{k,h})$ gives

$$\Delta_{k,h}(s_{k,h}) = V_{\star,h}(s_{k,h}) - Q_{\pi_k,h}(s_{k,h}, a_{k,h})$$
$$= \big(V_{\star,h}(s_{k,h}) - Q_{\star,h}(s_{k,h}, a_{k,h})\big) + \big(Q_{\star,h}(s_{k,h}, a_{k,h}) - Q_{\pi_k,h}(s_{k,h}, a_{k,h})\big)$$
$$= \underbrace{V_{\star,h}(s_{k,h}) - Q_{\star,h}(s_{k,h}, a_{k,h})}_{\text{action suboptimality at } (s_{k,h}, h)} + \underbrace{Q_{\star,h}(s_{k,h}, a_{k,h}) - Q_{\pi_k,h}(s_{k,h}, a_{k,h})}_{\text{future value gap at } (s_{k,h}, a_{k,h}, h)}.$$

By the Bellman equations and the derivation above, the second term equals

$$Q_{\star,h}(s_{k,h}, a_{k,h}) - Q_{\pi_k,h}(s_{k,h}, a_{k,h}) = \mathbb{E}\big[\Delta_{k,h+1}(s_{k,h+1}) \mid s_{k,h}, a_{k,h}\big],$$

which shows that the future value gap at step $h$ is exactly the expected value-difference at the next step. To bound the action–suboptimality term $V_{\star,h}(s_{k,h}) - Q_{\star,h}(s_{k,h}, a_{k,h})$, let $a_h^\star$ denote an $h$–optimal action at state $s_{k,h}$, so that

$$V_{\star,h}(s_{k,h}) = Q_{\star,h}(s_{k,h}, a_h^\star).$$

We apply the optimism inequality from Definition 5 to both $(s_{k,h}, a_h^\star)$ and $(s_{k,h}, a_{k,h})$. For the optimal action,

$$Q_{\star,h}(s_{k,h}, a_h^\star) \leq \widetilde{Q}_{t_{k,h}}(s_{k,h}, a_h^\star) + \text{bias}_{s_{k,h}, a_h^\star, t_{k,h}}.$$

For the chosen action we use the lower bound from Lemma 4, which gives

$$Q_{\star,h}(s_{k,h}, a_{k,h}) \geq \widetilde{Q}_{t_{k,h}}(s_{k,h}, a_{k,h}) - \beta_{t_{k,h}} \sqrt{v_{s_{k,h}, a_{k,h}, t_{k,h}}^{\text{post}}} - \text{bias}_{s_{k,h}, a_{k,h}, t_{k,h}}.$$

Because SPICE chooses $a_{k,h}$ greedily with respect to $\widetilde{Q}_{t_{k,h}}(s_{k,h}, \cdot)$, we have

$$\widetilde{Q}_{t_{k,h}}(s_{k,h}, a_h^\star) \leq \widetilde{Q}_{t_{k,h}}(s_{k,h}, a_{k,h}).$$

Combining these inequalities yields

$$V_{\star,h}(s_{k,h}) - Q_{\star,h}(s_{k,h}, a_{k,h}) = Q_{\star,h}(s_{k,h}, a_h^\star) - Q_{\star,h}(s_{k,h}, a_{k,h})$$

$$\leq \widetilde{Q}_{t_{k,h}}(s_{k,h}, a_{k,h}) + \text{bias}_{s_{k,h}, a_h^\star, t_{k,h}}$$
$$- \Big(\widetilde{Q}_{t_{k,h}}(s_{k,h}, a_{k,h}) - \beta_{t_{k,h}} \sqrt{v_{s_{k,h}, a_{k,h}, t_{k,h}}^{\text{post}}} - \text{bias}_{s_{k,h}, a_{k,h}, t_{k,h}}\Big)$$

$$\leq \beta_{t_{k,h}} \sqrt{v_{s_{k,h}, a_{k,h}, t_{k,h}}^{\text{post}}} + \text{bias}_{s_{k,h}, a_h^\star, t_{k,h}} + \text{bias}_{s_{k,h}, a_{k,h}, t_{k,h}}.$$

Combining the decomposition of $\Delta_{k,h}(s_{k,h})$ with the bound on the action–suboptimality term, we obtain, on the high–probability event of Lemma 4,

$$\Delta_{k,h}(s_{k,h}) = \big(V_{\star,h}(s_{k,h}) - Q_{\star,h}(s_{k,h}, a_{k,h})\big) + \big(Q_{\star,h}(s_{k,h}, a_{k,h}) - Q_{\pi_k,h}(s_{k,h}, a_{k,h})\big)$$

$$\leq \beta_{t_{k,h}} \sqrt{v_{s_{k,h}, a_{k,h}, t_{k,h}}^{\text{post}}} + \text{bias}_{s_{k,h}, a_h^\star, t_{k,h}} + \text{bias}_{s_{k,h}, a_{k,h}, t_{k,h}} + \mathbb{E}\big[\Delta_{k,h+1}(s_{k,h+1}) \mid s_{k,h}, a_{k,h}\big].$$

Since

$$\text{bias}_{s,a,t} = \frac{N_{s,a}^{\text{pri}}}{N_{s,a}^{\text{pri}} + N_{s,a,t}} \big|\mu_{s,a}^{\text{pri}} - Q_\star(s, a)\big| \leq \big|\mu_{s,a}^{\text{pri}} - Q_\star(s, a)\big|,$$

each bias term is uniformly bounded. Let

$$B_{\max} := \max_{(s,a)} \big|\mu_{s,a}^{\text{pri}} - Q_\star(s, a)\big| < \infty,$$

so that $\text{bias}_{s,a^\star,t} \leq B_{\max}$ for all $t$. Moreover, $\beta_t \sqrt{v_{s,a,t}^{\text{post}}} + \text{bias}_{s,a,t} \geq \text{bias}_{s,a,t} \geq 0$. Thus we may choose a constant $C_b \geq B_{\max}$ such that, for every $(s, a, t)$,

$$\text{bias}_{s,a^\star,t} \leq C_b\big(\beta_t \sqrt{v_{s,a,t}^{\text{post}}} + \text{bias}_{s,a,t}\big).$$

This allows us to absorb the optimal-action bias into a single multiplicative factor on the bonus terms. Absorbing $\text{bias}_{s_{k,h}, a_h^\star, t_{k,h}}$ into a multiplicative constant in front of $\beta_{t_{k,h}} \sqrt{v_{s_{k,h}, a_{k,h}, t_{k,h}}^{\text{post}}} + \text{bias}_{s_{k,h}, a_{k,h}, t_{k,h}}$, we conclude that there exists a constant $C_0 \geq 1$ such that

$$\Delta_{k,h}(s_{k,h}) \leq C_0\Big(\beta_{t_{k,h}} \sqrt{v_{s_{k,h}, a_{k,h}, t_{k,h}}^{\text{post}}} + \text{bias}_{s_{k,h}, a_{k,h}, t_{k,h}}\Big) + \mathbb{E}\big[\Delta_{k,h+1}(s_{k,h+1}) \mid s_{k,h}, a_{k,h}\big].$$

$$(37)$$

We take expectations with respect to the randomness in episode $k$ and define
$$\delta_{k,h} := \mathbb{E}[\Delta_{k,h}(s_{k,h})].$$

Applying expectations to equation 37 and using the law of total expectation gives
$$\delta_{k,h} \leq C_0 \, \mathbb{E}\Big[\beta_{t_{k,h}} \sqrt{v^{\text{post}}_{s_{k,h},a_{k,h},t_{k,h}}} + \text{bias}_{s_{k,h},a_{k,h},t_{k,h}}\Big] + \mathbb{E}[\mathbb{E}[\Delta_{k,h+1}(s_{k,h+1}) \mid s_{k,h}, a_{k,h}]]$$
$$= C_0 \, \mathbb{E}\Big[\beta_{t_{k,h}} \sqrt{v^{\text{post}}_{s_{k,h},a_{k,h},t_{k,h}}} + \text{bias}_{s_{k,h},a_{k,h},t_{k,h}}\Big] + \delta_{k,h+1}.$$

At step $H+1$ the episode terminates, so $\Delta_{k,H+1}(s) = 0$ and hence $\delta_{k,H+1} = 0$. Starting from the one-step recursion
$$\delta_{k,h} \leq C_0 \, \mathbb{E}\Big[\beta_{t_{k,h}} \sqrt{v^{\text{post}}_{s_{k,h},a_{k,h},t_{k,h}}} + \text{bias}_{s_{k,h},a_{k,h},t_{k,h}}\Big] + \delta_{k,h+1},$$

and using the terminal condition $\delta_{k,H+1} = 0$, we expand the first few steps:
$$\delta_{k,H} \leq C_0 \, \mathbb{E}[\text{bonus}_{k,H}],$$
$$\delta_{k,H-1} \leq C_0 \, \mathbb{E}[\text{bonus}_{k,H-1}] + \delta_{k,H} \leq C_0 \left(\mathbb{E}[\text{bonus}_{k,H-1}] + \mathbb{E}[\text{bonus}_{k,H}]\right),$$
and similarly,
$$\delta_{k,H-2} \leq C_0 \left(\mathbb{E}[\text{bonus}_{k,H-2}] + \mathbb{E}[\text{bonus}_{k,H-1}] + \mathbb{E}[\text{bonus}_{k,H}]\right).$$

By iterating the recursion $\delta_{k,h} \leq C_0 \, \mathbb{E}[\text{bonus}_{k,h}] + \delta_{k,h+1}$ backwards from $h = H$ using the terminal condition $\delta_{k,H+1} = 0$, one easily checks by induction that
$$\delta_{k,h} \leq C_0 \sum_{j=h}^{H} \mathbb{E}[\text{bonus}_{k,j}] \quad \text{for all } h.$$

In particular, for $h = 1$ this yields
$$\delta_{k,1} \leq C_0 \sum_{h=1}^{H} \mathbb{E}[\text{bonus}_{k,h}],$$

as claimed. where
$$\text{bonus}_{k,h} = \beta_{t_{k,h}} \sqrt{v^{\text{post}}_{s_{k,h},a_{k,h},t_{k,h}}} + \text{bias}_{s_{k,h},a_{k,h},t_{k,h}}.$$

This gives
$$\delta_{k,1} \leq C_0 \sum_{h=1}^{H} \mathbb{E}\Big[\beta_{t_{k,h}} \sqrt{v^{\text{post}}_{s_{k,h},a_{k,h},t_{k,h}}} + \text{bias}_{s_{k,h},a_{k,h},t_{k,h}}\Big].$$

Recalling that $\delta_{k,1} = \mathbb{E}\big[V_\star(s_1^k) - V_{\pi_k}(s_1^k)\big]$, we obtain the per–episode regret bound
$$\mathbb{E}\big[V_\star(s_1^k) - V_{\pi_k}(s_1^k)\big] \leq C_0 \sum_{h=1}^{H} \mathbb{E}\Big[\beta_{t_{k,h}} \sqrt{v^{\text{post}}_{s_{k,h},a_{k,h},t_{k,h}}} + \text{bias}_{s_{k,h},a_{k,h},t_{k,h}}\Big]. \tag{38}$$

This completes the proof of Lemma 5. $\qquad\square$

We now sum the estimation-error part over all episodes. We define the cumulative (global) regret
$$\text{Regret}_K := \sum_{k=1}^{K} \big(V_\star(s_1^k) - V_{\pi_k}(s_1^k)\big).$$

Conditioned on the high-probability event of Lemma 4 and summing equation 38 over episodes $k = 1, \ldots, K$ gives
$$\text{Regret}_K \leq C_0 \sum_{k=1}^{K} \sum_{h=1}^{H} \beta_{t_{k,h}} \sqrt{v^{\text{post}}_{s_{k,h},a_{k,h},t_{k,h}}} + C_0 \sum_{k=1}^{K} \sum_{h=1}^{H} \text{bias}_{s_{k,h},a_{k,h},t_{k,h}}. \tag{39}$$

We treat the two sums separately: the first captures statistical estimation error, the second captures the warm-start effect due to prior miscalibration.

**Lemma 6** (Bounding the UCB bonus term.). *There exists a constant $C_1 > 0$ such that*

$$\sum_{k=1}^{K}\sum_{h=1}^{H} \beta_{t_{k,h}} \sqrt{v_{s_{k,h},a_{k,h},t_{k,h}}^{\text{post}}} \;\leq\; C_1 \, H \, \sqrt{SAK}$$

*on the high-probability event of Lemma 4.*

*Proof.* We recall that the posterior variance for $(s,a)$ at time $t$ satisfies

$$v_{s,a,t}^{\text{post}} = \frac{\sigma_Q^2}{N_{s,a}^{\text{pri}} + N_{s,a,t}},$$

where $N_{s,a,t}$ is the visit count to $(s,a)$ up to time $t$ and $\sigma_Q^2$ is the sub-Gaussian variance proxy from equation 29. Since $\beta_t$ is non-decreasing in $t$ and of order $\Theta(\sqrt{\log(SAT)})$, we can upper bound each $\beta_{t_{k,h}}$ by $\beta := \beta_T$ (a constant factor depending only on $S, A, T$ and the prior), so that

$$\sum_{k=1}^{K}\sum_{h=1}^{H} \beta_{t_{k,h}} \sqrt{v_{s_{k,h},a_{k,h},t_{k,h}}^{\text{post}}} \;\leq\; \beta \sum_{k=1}^{K}\sum_{h=1}^{H} \sqrt{v_{s_{k,h},a_{k,h},t_{k,h}}^{\text{post}}}.$$

We now group visits by state–action pair. Let $N_{s,a}$ be the total number of visits to $(s,a)$ over all $KH$ steps, so that

$$N_{s,a} := \sum_{k=1}^{K}\sum_{h=1}^{H} \mathbf{1}\{s_{k,h} = s,\, a_{k,h} = a\}, \qquad \sum_{(s,a)} N_{s,a} = KH.$$

For the $j$-th visit to $(s,a)$ we have $v_{s,a,\cdot}^{\text{post}} = \sigma_Q^2/(N_{s,a}^{\text{pri}} + j - 1)$, so

$$\sum_{k=1}^{K}\sum_{h=1}^{H} \sqrt{v_{s_{k,h},a_{k,h},t_{k,h}}^{\text{post}}} = \sum_{(s,a)}\sum_{j=1}^{N_{s,a}} \sqrt{\frac{\sigma_Q^2}{N_{s,a}^{\text{pri}} + j - 1}}$$

$$= \sum_{(s,a)}\sum_{j=1}^{N_{s,a}} \frac{\sigma_Q}{\sqrt{N_{s,a}^{\text{pri}} + j - 1}}$$

$$\leq \sum_{(s,a)}\sum_{j=1}^{N_{s,a}} \frac{\sigma_Q}{\sqrt{j}} \qquad (\text{since } N_{s,a}^{\text{pri}} \geq 1)$$

$$= \sigma_Q \sum_{(s,a)}\sum_{j=1}^{N_{s,a}} \frac{1}{\sqrt{j}}$$

$$\leq \sigma_Q \sum_{(s,a)} \left( \sum_{j=1}^{N_{s,a}} \frac{1}{\sqrt{j}} \right)$$

$$\leq \sigma_Q \sum_{(s,a)} \left( \sum_{j=1}^{N_{s,a}} \frac{1}{\sqrt{j}} \right)$$

$$\leq \sigma_Q \sum_{(s,a)} \left( 1 + \int_{1}^{N_{s,a}} x^{-1/2}\, dx \right) \qquad (\text{integral comparison for decreasing } x^{-1/2})$$

$$= \sigma_Q \sum_{(s,a)} \left( 1 + 2(\sqrt{N_{s,a}} - 1) \right)$$

$$\leq 2\sigma_Q \sum_{(s,a)} \sqrt{N_{s,a}} \qquad (\text{since } 1 + 2(\sqrt{N_{s,a}} - 1) \leq 2\sqrt{N_{s,a}})$$

$$= 2\sigma_Q \sum_{(s,a)} \sqrt{N_{s,a}}.$$

Applying the Cauchy–Schwarz inequality with $u_{s,a} = 1$ and $v_{s,a} = \sqrt{N_{s,a}}$, we get

$$\sum_{(s,a)} \sqrt{N_{s,a}} = \sum_{(s,a)} u_{s,a} v_{s,a} \leq \Big(\sum_{(s,a)} u_{s,a}^2\Big)^{1/2} \Big(\sum_{(s,a)} v_{s,a}^2\Big)^{1/2}.$$

Since $\sum_{(s,a)} u_{s,a}^2 = SA$ and $\sum_{(s,a)} v_{s,a}^2 = \sum_{(s,a)} N_{s,a} = KH$, this becomes

$$\sum_{(s,a)} \sqrt{N_{s,a}} \leq \sqrt{SA}\sqrt{KH} = \sqrt{SA \cdot KH}.$$

From the previous two inequalities we have

$$\sum_{k=1}^{K} \sum_{h=1}^{H} \sqrt{v_{s_{k,h}, a_{k,h}, t_{k,h}}^{\text{post}}} \leq 2\sigma_Q \sum_{(s,a)} \sqrt{N_{s,a}}$$

$$\leq 2\sigma_Q \sqrt{SA \cdot KH}$$

From the previous inequalities we have

$$\sum_{k=1}^{K} \sum_{h=1}^{H} \sqrt{v_{s_{k,h}, a_{k,h}, t_{k,h}}^{\text{post}}} \leq 2\sigma_Q \sqrt{SA \cdot KH}.$$

Therefore,

$$\sum_{k=1}^{K} \sum_{h=1}^{H} \beta_{t_{k,h}} \sqrt{v_{s_{k,h}, a_{k,h}, t_{k,h}}^{\text{post}}} \leq \beta \cdot 2\sigma_Q \sqrt{SA \cdot KH},$$

where $\beta := \beta_T$. By the assumption in Eq. equation 29 that $\sigma_Q^2 \leq c_H H$, we have $\sigma_Q \leq \sqrt{c_H H}$, and hence

$$\beta \cdot 2\sigma_Q \sqrt{SA \cdot KH} \leq 2\beta \sqrt{c_H H} \sqrt{SA \cdot KH} = 2\beta \sqrt{c_H}\, H \sqrt{SAK}.$$

Defining $C_1 := 2\beta\sqrt{c_H}$, we obtain

$$\sum_{k=1}^{K} \sum_{h=1}^{H} \beta_{t_{k,h}} \sqrt{v_{s_{k,h}, a_{k,h}, t_{k,h}}^{\text{post}}} \leq C_1\, H \sqrt{SAK},$$

as claimed. $\qquad\square$

Thus the first term on the right-hand side of equation 39 contributes $O\big(H\sqrt{SAK}\big)$ to the cumulative regret (up to logarithmic factors absorbed into $C_1$).

**Lemma 7** (Bounding the warm-start term.). *On the high–probability event of Lemma 4, the cumulative contribution of the prior-bias terms satisfies*

$$\sum_{k=1}^{K} \sum_{h=1}^{H} \text{bias}_{s_{k,h}, a_{k,h}, t_{k,h}} \leq \sum_{(s,a) \in \mathcal{S} \times \mathcal{A}} O\Big(N_{s,a}^{\text{pri}}\, \big|\mu_{s,a}^{\text{pri}} - Q_\star(s,a)\big|\Big).$$

*Thus, any prior miscalibration only contributes an additive, state–action–wise constant to the regret.*

*Proof.* We fix a state–action pair $(s, a)$ and consider the sequence of global times at which $(s, a)$ is visited:

$$\tau_1(s,a) < \tau_2(s,a) < \cdots < \tau_{N_{s,a}}(s,a).$$

At the $j$-th visit, the number of previous visits to $(s,a)$ is $N_{s,a,\tau_j(s,a)} = j - 1$, so the corresponding bias term equals

$$\text{bias}_{s,a,\tau_j(s,a)} = \frac{N_{s,a}^{\text{pri}}}{N_{s,a}^{\text{pri}} + j - 1}\left|\mu_{s,a}^{\text{pri}} - Q_\star(s,a)\right|.$$

The total contribution of $(s,a)$ to the warm-start sum is therefore

$$\sum_{j=1}^{N_{s,a}} \text{bias}_{s,a,\tau_j(s,a)} = \left|\mu_{s,a}^{\text{pri}} - Q_\star(s,a)\right| \sum_{j=0}^{N_{s,a}-1} \frac{N_{s,a}^{\text{pri}}}{N_{s,a}^{\text{pri}} + j}.$$

We now bound the inner sum deterministically. For any integer $N \geq 1$,

$$\sum_{j=0}^{N-1} \frac{N_{s,a}^{\text{pri}}}{N_{s,a}^{\text{pri}} + j} \leq N_{s,a}^{\text{pri}} \int_0^N \frac{1}{N_{s,a}^{\text{pri}} + x}\, dx = N_{s,a}^{\text{pri}}\left[\log(N_{s,a}^{\text{pri}} + N) - \log N_{s,a}^{\text{pri}}\right].$$

Using the identity

$$\log\left(N_{s,a}^{\text{pri}} + N\right) - \log\left(N_{s,a}^{\text{pri}}\right) = \log\left(1 + \frac{N}{N_{s,a}^{\text{pri}}}\right)$$

and the bound $\log(1 + x) \leq x$ for all $x \geq 0$, we obtain

$$\sum_{j=0}^{N-1} \frac{N_{s,a}^{\text{pri}}}{N_{s,a}^{\text{pri}} + j} \leq N_{s,a}^{\text{pri}} \log\left(1 + \frac{N}{N_{s,a}^{\text{pri}}}\right) \leq N_{s,a}^{\text{pri}} \frac{N}{N_{s,a}^{\text{pri}}} = N.$$

Since the number of visits to $(s,a)$ cannot exceed the total number of interaction steps, we have $N \leq T = KH$. Combining this with the previous bound gives

$$\sum_{j=0}^{N-1} \frac{N_{s,a}^{\text{pri}}}{N_{s,a}^{\text{pri}} + j} \leq N \leq KH.$$

Because $KH$ is a fixed problem-dependent constant and $N_{s,a}^{\text{pri}} \geq 1$, we may rewrite this as

$$N \leq \frac{KH}{N_{s,a}^{\text{pri}}} N_{s,a}^{\text{pri}} = c_{s,a} N_{s,a}^{\text{pri}},$$

where $c_{s,a} := KH/N_{s,a}^{\text{pri}}$ is a finite constant depending only on the prior and the horizon. Absorbing $c_{s,a}$ into big-$O$ notation gives the desired bound

$$\sum_{j=0}^{N-1} \frac{N_{s,a}^{\text{pri}}}{N_{s,a}^{\text{pri}} + j} \leq O\left(N_{s,a}^{\text{pri}}\right).$$

Note that the bound on $\sum_{j=1}^{N_{s,a}} \text{bias}_{s,a,\tau_j(s,a)}$ depends only on the number of visits $N_{s,a}$ and the prior pseudo-count $N_{s,a}^{\text{pri}}$, and not on the particular order in which visits to $(s,a)$ occur. In other words, the interleaving of $(s,a)$ with visits to other state–action pairs plays no role. Combining the expression for the warm-start sum with the logarithmic bound obtained above yields, for each $(s,a)$,

$$\sum_{j=1}^{N_{s,a}} \text{bias}_{s,a,\tau_j(s,a)} \leq c\, N_{s,a}^{\text{pri}}\left|\mu_{s,a}^{\text{pri}} - Q_\star(s,a)\right|,$$

for a constant $c > 0$. Summing over all $(s,a)$ gives

$$\sum_{k=1}^{K}\sum_{h=1}^{H} \text{bias}_{s_{k,h},a_{k,h},t_{k,h}} \leq \sum_{(s,a)} O\left(N_{s,a}^{\text{pri}}\left|\mu_{s,a}^{\text{pri}} - Q_\star(s,a)\right|\right),$$

which provides the desired warm-start contribution. $\qquad\square$

Combining Lemma 6 and Lemma 7 with equation 39, we obtain on the event of Lemma 4,

$$\text{Regret}_K \leq O(H\sqrt{SAK}) + \sum_{(s,a)} O\Big(N_{s,a}^{\text{pri}} \big|\mu_{s,a}^{\text{pri}} - Q_\star(s,a)\big|\Big).$$

The failure event of Lemma 4 has probability $O(1/K)$, and in the worst case each episode contributes at most $H$ regret, so its contribution to the expected regret is at most $O(1)$. Taking expectations on both sides and absorbing this constant into the big-$O$ terms yields exactly the bound in Eq. equation 28, completing the proof of Theorem 2.

## D  ADDITIONAL EXPERIMENTAL RESULTS

### D.1  BANDIT SETTING

**Setup and data.**  Each task is a stochastic $A$-armed bandit with i.i.d. arm means $\mu_a \sim \text{Unif}[0,1]$ and Gaussian rewards $r \sim \mathcal{N}(\mu_a, \sigma^2)$. Unless noted, $A{=}5$, horizon $H{=}500$, and the default test noise is $\sigma{=}0.3$. We evaluate on $N{=}200$ held-out environments; for robustness we fix the means and sweep $\sigma \in \{0.0, 0.3, 0.5\}$ at test time. For SPICE/DPT we report the mean over 3 seeds. Offline we measure suboptimality $\mu^\star - \mu_{\hat{a}}$ as a function of context length $h$; online we report cumulative regret $\sum_{t=1}^{H}(\mu^\star - \mu_{a_t})$.

### D.2  ABLATION: QUALITY OF TRAINING DATA

We use weakmix80 as a less-poor dataset: labels are $80\%$ optimal and the contexts remain heterogeneous due to the mixed-random behaviour.

**Setup.**  Each task is a stochastic $A$-armed bandit with i.i.d. arm means $\mu_a \sim \text{Unif}[0,1]$ and rewards $r \sim \mathcal{N}(\mu_a, \sigma^2)$. We use $A{=}20$, horizon $H{=}500$, and default test noise $\sigma{=}0.3$. We evaluate on $N{=}200$ held-out environments.

**Data generation (weakmix80).**  Following the DPT protocol, contexts are collected by a behaviour policy that mixes broad exploration with concentrated exploitation on one arm. Concretely, for each environment we form a per-arm distribution

$$p = (1-\omega)\,\text{Dirichlet}(\mathbf{1}) + \omega\,\delta_{i^\star},$$

where $\delta_{i^\star}$ is a point mass on a single arm $i^\star$ (chosen uniformly at random for this experiment), and we fix the mix strength to $\omega{=}0.5$. At each context step an action is drawn from $p$. Supervision is *weak*: the training label is generated in `mix` mode with probability $q{=}0.8$ using the true optimal arm, and with probability $1-q$ by sampling an arm from $p$. We denote this setting by **weakmix80**. We generate 100k training tasks and 200 evaluation tasks with the above roll-in and labels.

**Models and deployment.**  DPT and SPICE share the same transformer trunk (6 layers, 64 hidden units, single head, no dropout). For this ablation we pretrain both for 100 epochs on the weakmix80 dataset with $A{=}20$. Offline, all methods select a single arm from a fixed context. Online, they interact for $H$ steps starting from an empty context; SPICE acts with a posterior-UCB controller, DPT samples from its predicted action distribution, and classical bandits (Emp, UCB, TS) use standard update rules.

**Results.**

- **Offline .** DPT is competitive offline under weakmix80 (80% optimal labels), but still converges more slowly than TS/SPICE as $h$ grows (Fig. 6a).

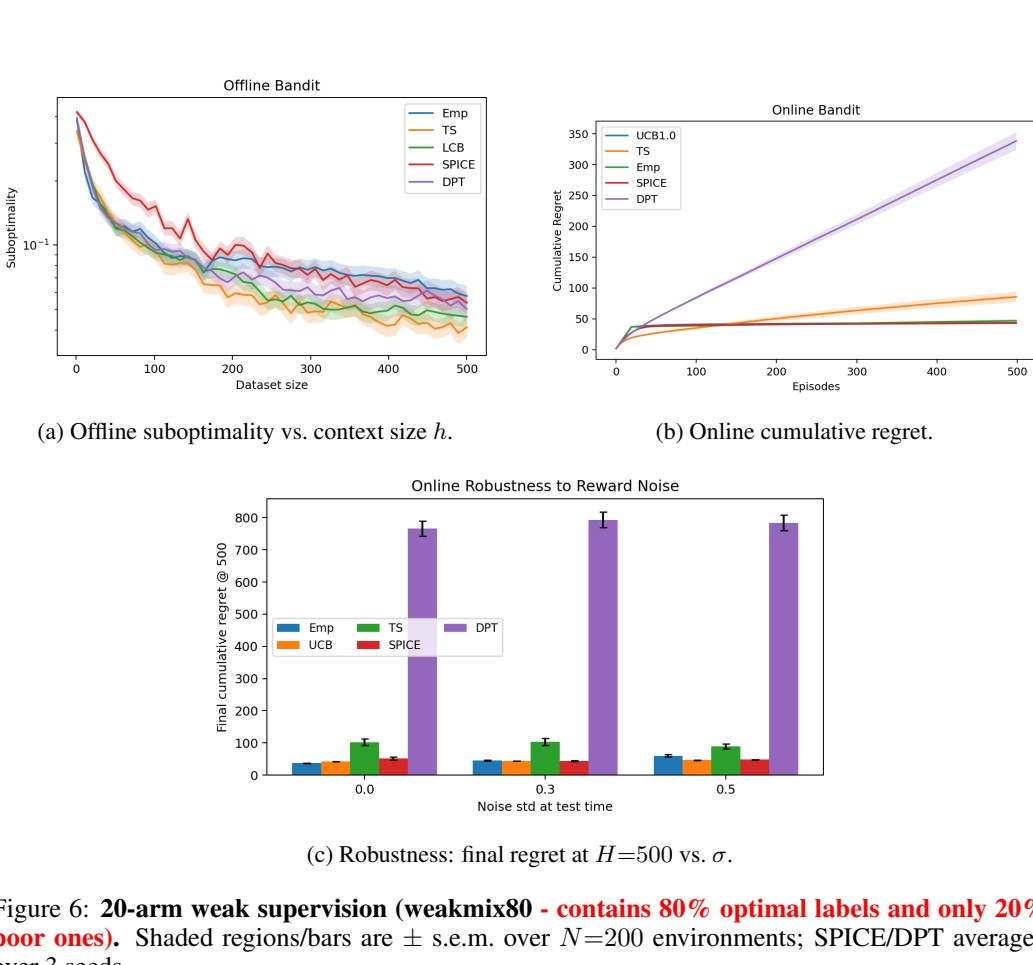

(a) Offline suboptimality vs. context size $h$.

(b) Online cumulative regret.

(c) Robustness: final regret at $H{=}500$ vs. $\sigma$.

Figure 6: **20-arm weak supervision (weakmix80 - contains 80% optimal labels and only 20% poor ones).** Shaded regions/bars are $\pm$ s.e.m. over $N{=}200$ environments; SPICE/DPT averaged over 3 seeds.

- **Online .** SPICE attains the lowest regret among learned methods and closely tracks UCB, while TS is slightly worse and Emp is clearly worse (Fig. 6b). In contrast, DPT exhibits near-linear growth in regret: it improves little with additional interaction despite 80% optimal labels.

- **Robustness to reward-noise shift.** SPICE, TS, UCB and Emp degrade smoothly as $\sigma$ increases, with small absolute changes. DPT's final regret remains orders of magnitude larger is and essentially insensitive to $\sigma$, indicating failure to adapt from weak training data (Fig. 6c).

### D.3 ABLATION: WEIGHT TERMS IN TRAINING OBJECTIVE

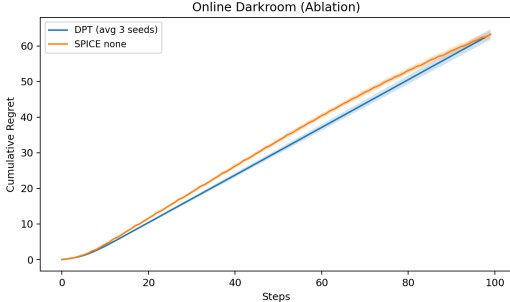

Figure 7: Online Darkroom ablation on weighting terms. We compare DPT against SPICE trained with no weights in the pretraining objective (both averaged over 3 seeds).

In this ablation, we studied the effect of the weighting terms in the SPICE objective (importance, advantage, epistemic). The results show that removing these terms degrades performance, highlighting their role in shaping better representations under weak data and reducing online regret.

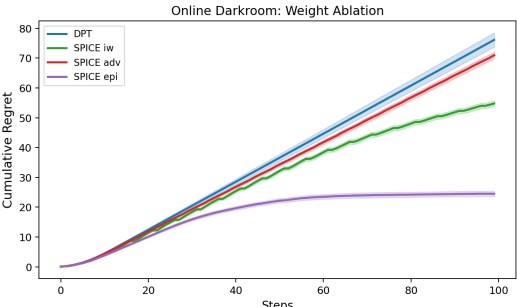

Figure 8: Weight ablation on the Darkroom environment. We compare DPT against SPICE variants trained with individual weighting components (iw: importance weighting, adv: advantage weighting, or epi: epistemic weighting) to isolate the contribution of each term. Results are averaged over 3 seeds.

We conduct a weight factor ablation study on the Darkroom environment to understand the individual contributions of each weighting component in the SPICE training objective. Our results demonstrate contributions from each weighting mechanism. Epistemic weighting achieves the strongest performance, achieving a cumulative regret of approximately 24 by step 60 and maintaining this level through step 100, representing a 68% reduction compared to the DPT baseline (76 cumulative regret). This suggests that prioritising uncertain actions, where the Q-head ensemble shows high disagreement, is particularly effective for exploration in this sparse-reward setting. Importance weighting provides moderate improvement, reducing cumulative regret to 54 at step 100 (29% reduction), indicating that correcting for distribution shift between the behaviour policy and uniform target policy gives meaningful benefits. Advantage weighting also shows improvement, achieving 71 cumulative regret (7% reduction), demonstrating that emphasising high-advantage transitions alone is insufficient for this task. The combination of all three weights shown in Figure 4b achieves the lowest regret. These findings highlight that epistemic uncertainty-based weighting is the primary driver of SPICE's performance in the Darkroom environment, with importance weighting and advantage weighting providing additional performance benefits.

# E    USE OF LLMS.

ChatGPT was employed as a general-purpose assistant for enhancing writing clarity, conciseness, and tone, and providing technical coding support for plotting utilities and minor debugging tasks. All outputs were verified by the authors, who retain full responsibility for research conception, algorithmic contributions, implementation, experimental findings, and manuscript writing.

# F    ETHICS STATEMENT.

All authors have read and adhere to the ICLR Code of Ethics. This work does not involve human subjects, personally identifiable data, or sensitive attributes. We evaluate solely on synthetic bandit and control benchmarks and do not deploy in safety-critical settings. We discuss limitations (kernel choice, sub-Gaussian noise assumption, misspecified priors) and avoid claims beyond our experimental scope (Section 7). Our method could, in principle, be applied to high-stakes domains; we therefore emphasise the need for rigorous safety evaluation and domain-appropriate oversight before any real-world use.

