# OpenReview forum: "In-Context Reinforcement Learning through Bayesian Fusion of Context and Value Prior"
_ICLR.cc/2026/Conference — Submitted to ICLR 2026_

### Official Review · Reviewer_SFAi · 2025-10-30

**Soundness:** 3
**Presentation:** 3
**Contribution:** 3
**Rating:** 8
**Confidence:** 3

**Summary:**

The paper proposes a Bayesian ICRL method named SPICE that enables efficient adaptation from suboptimal offline data. This method introduces a value-function ensemble trained with Bayesian shrinkage to learn a calibrated prior over action values, and performs Bayesian fusion at inference time to combine this prior with task-specific contextual evidence. SPICE further employs a weighted policy-head objective to improve representation learning under behavior-policy bias. The authors provide a theoretical analysis in the stochastic bandit setting, establishing a logarithmic regret bound for the proposed posterior-UCB controller, and demonstrate through experiments on both bandit and MDP environments (e.g., Dark Room) that SPICE achieves significantly lower cumulative regret and higher returns compared to existing in-context RL baselines.

**Strengths:**

1. The authors integrate a policy head and value ensemble head to the Transformer backbone
    1. In policy head, the authors include three different complementary weights to mitigate behavior policy bias, emphasize reward-relevant behaviors, and focus learning on uncertain or under-explored regions, respectively. These mechanisms enhance representation learning from suboptimal offline data and improve the backbone model's ability to support reliable value estimation.
    1. In value ensemble head, the authors employ Bayesian shrinkage during training to obtain a calibrated value prior, and perform Bayesian fusion at inference to combine this prior with task-specific contextual evidence. These design choices enable principled uncertainty modeling, task generalization, and coherent exploration under limited or noisy data.
1. In addition to the architecture innovations, this paper also gives a rigorous theoretical analysis of the proposed Bayesian controller in the stochastic bandit setting, providing formula bounds of regret demonstrating how prior miscalibration only introduces a constant warm-start penalty. This analysis offers a sound theoretical foundation for Bayesian ICRL.
1. The empirical experiments well match their theory, showing both the correctness of the theory contribution and the effectiveness in the real tasks.

**Weaknesses:**

1. While ICRL methods are often tested in simplified environments such as Dark Room, it would strengthen the paper to evaluate SPICE on high-dimensional continuous control benchmarks (e.g., MuJoCo or Meta-World). These are widely adopted in reinforcement learning research and would better demonstrate the method’s scalability and generalization beyond discrete or low-dimensional tasks.
1. The choice of PPO as a baseline may be suboptimal, as PPO is an on-policy algorithm that is typically sample-inefficient and struggles in sparse-reward settings. Using off-policy algorithms (e.g., DDPG, SAC) or PPO combined with Hindsight Experience Replay (HER) [1] would provide a fairer and more competitive comparison, particularly for the sparse-reward nature of the Dark Room environment.
1. The paper does not discuss the choice of kernel function used for measuring context-state similarity during Bayesian fusion, even though this choice can affect the final performance. A discussion or ablation on kernel type would clarify how robust the method is to this critical design decision.

[1] Crowder, Douglas C., et al. "Hindsight experience replay accelerates proximal policy optimization." arXiv preprint arXiv:2410.22524 (2024).

**Questions:**

1. Could the authors extend regret or sample-complexity analysis to MDPs? This would significantly strengthen the theoretical contribution and demonstrate the broader applicability of SPICE beyond the bandit setting.
1. While the three weighting factors in the policy head intuitively improve representation learning from suboptimal offline data, the policy head itself is not used during inference. Could the authors quantify how much these weighting schemes affect the overall performance and how sensitive is the algorithm to the policy-head training objective and the associated hyperparameters?

---

> ### Author Response · Authors · 2025-11-20
>
> We very much appreciate the thoughtful review and recognition of the architectural novelty (ensemble, weighted supervision) and the rigorous theoretical foundation of SPICE. We are grateful for the suggestions to add a theoretical analysis for MDPs, as well as to expand the evaluation to a high-dimensional environment and to further expand the weighting factors ablation.
>
> ## What we changed since the initial version
>
> 1. High-dimensional benchmark: We are currently performing experiments to evaluate SPICE's performance and robustness on Miniworld, which is a more challenging environment where the agent receives RGB image observations. This environment is commonly used in the ICRL literature and will serve as a good testbed for evaluating the generalisability of SPICE's learned policies.
> 2. PPO baseline: PPO is used as a standard single-task RL reference for sample-efficiency in the ICLR literature, but we acknowledge the usefulness of including even more comparisons. We will include DDPG and SAC in the experiments that we are currently running.
> 3. Kernel discussion: We have added more details on kernel usage in Section 3.3, clarifying that SPICE applies the kernel to the learned latent feature space $H$ generated by the Transformer trunk, not to the raw state space $S$. The feature space is shaped using the weighted supervision objective and maps states with similar action values, advantage estimates, and epistemic uncertainty into nearby regions, making $H$ suitable for Bayesian fusion. We have also included practical guidance for the choice of kernel to Appendix A.7. In addition, we are currently running an ablation on the choice of kernel and bandwidth on the Darkroom and Miniworld environments.
>
> ## Answers to your questions
>
> Q1: Could the authors extend regret or sample-complexity analysis to MDPs?
>
> Thank you for this good suggestion, we have added Theorem 2 (Sec. 4.2) to extend the theory to finite-horizon MDPs and confirm that SPICE retains an optimal $O(H \sqrt{S A K} ​)$ optimal asymptotic rate. The full proof is currently being finalised and will be included in the next revision. At a high level, the proof follows this structure:
> - Optimism via posterior UCB: we show that the SPICE posterior forms an upper confidence bound on $Q^\ast_h(s,a)$
> up to a small prior-bias term that decays with visits.
> - UCBVI-style regret decomposition: By plugging these optimistic Q-estimates into the standard dynamic regret decomposition over horizon $H$, we obtain a regret bound matching UCBVI.
>
>
>
> Q2: While the three weighting factors in the policy head intuitively improve representation learning from suboptimal offline data, the policy head itself is not used during inference. Could the authors quantify how much these weighting schemes affect the overall performance and how sensitive is the algorithm to the policy-head training objective and the associated hyperparameters?
>
> Thank you for this question. Appendix C3 contains an ablation on removing all three weights and shows that SPICE performs much worse without the weight terms. We are currently running an additional ablation experiment to isolate the impact of each individual weight on SPICE’s performance.

---

> > ### Author Response · Authors · 2025-12-03
> >
> > Thank you for your very positive and detailed review, and for the concrete suggestions on theory and experiments.
> > - Miniworld and stronger baselines: To move beyond simplified environments, we have set up Miniworld experiments with image observations and are currently running them. In this setting we are also expanding the baseline suite beyond PPO to include stronger off‑policy algorithms such as SAC (and related continuous‑control baselines), so that SPICE is evaluated against even more RL methods in this higher‑dimensional regime. We are additionally preparing a kernel‑ablation study in this environment, varying kernel type and bandwidth on the learned latent features.
> > - Extended theory (proof added): We have fully developed the theory that was previously only sketched. The MDP regret bound (Theorem 2) is now proven in Appendix C by lifting the bandit posterior-concentration analysis to state–action pairs and plugging the resulting confidence sets into a UCBVI-style dynamic regret decomposition. The proof shows that SPICE’s posterior‑UCB controller achieves the optimal $O(H \sqrt{SAK})$ regret rate for finite‑horizon tabular MDPs, matching classical UCBVI, while any prior miscalibration appears only as an additive warm‑start term that does not scale with K.
> > - Weighting‑scheme ablation: Responding to your question about the impact of the policy‑head weighting schemes, we have expanded the ablation. In addition to the “no weights” variant, we now isolate each component (importance, advantage, epistemic) and show how each affects Darkroom performance. The results confirm that the weights substantially improve regret, with epistemic weighting being the main driver and the combination of all three weights performing best.

---

### Official Review · Reviewer_Yryg · 2025-10-30

**Soundness:** 2
**Presentation:** 3
**Contribution:** 2
**Rating:** 4
**Confidence:** 4

**Summary:**

The paper proposes SPICE, a Bayesian ICRL method that learns an ensemble-based prior over Q-values from suboptimal data and, at test time, fuses this prior with kernel-weighted context statistics to form per-action posteriors. The agent acts greedily offline or with a posterior-UCB rule online. Experiments on bandits and a darkroom MDP show strong adaptation and regret reduction versus DPT/AD under weak supervision.

**Strengths:**

1. Learning an explicit, calibrated value prior for ICRL  is a clear and novel idea.
2. Theoretical contribution: proves regret-optimality (O(log K)) in stochastic bandits without requiring optimal pretraining.
3. Empirical validation aligns with theory: SPICE tracks UCB’s logarithmic regret and adapts quickly from suboptimal logs, whereas sequence-only ICRL baselines remain tied to behavior policy

**Weaknesses:**

1. The assumption of SPICE is too strong to limit its application.  The Bayesian fusion uses kernel-weighted evidence, implicitly assuming  a form of local smoothness in the action-value landscape—i.e., that states close under the chosen kernel share similar action values, and consequently that evidence from nearby states is informative for the query. However, this can break in domains with discontinuous or highly multimodal action-value structure where small perturbations in state can correspond to very different optimal actions.  Thus it is hard to apply this method to more complcated real-world tasks, unless the authors prove that using experiments results.
2. The method assumes the ensemble yields reasonably calibrated priors. Overconfident or biased ensembles can harm short-horizon performance despite the bandit warm-start guarantee. No stress tests on deliberate miscalibration are provided
3. Validation is confined to bandits and a stylized darkroom MDP; scalability to complex, long-horizon, or partially observable tasks remains untested.
4. Comparisons omit recent in-context RL methods that do not require optimal labels (e.g., ICEE, DIT). Including them is important to discover the true value of this method
5. In Eq. (4), the policy uses three weighting factors (importance, advantage, epistemic). While advantage weighting is motivated for learning from suboptimal data, the contribution of the other two is less clear. A targeted ablation isolating each term would clarify their necessity and impact.

**Questions:**

1. Darkroom evaluation: Are the test tasks interpolation within the training task distribution or extrapolation  to out-of-distribution goals/dynamics? Please clarify how held-out tasks differ from training  and quantify the distributional shift.
2. Kernel sensitivity: How sensitive is performance to the choice of kernel and bandwidth in Bayesian fusion?

---

> ### Author Response · Authors · 2025-11-20
>
> We appreciate your thoughtful and detailed review and acknowledgment of the novelty (explicit, calibrated prior) and the theoretical strength of SPICE. Your suggestions for improvement are highly valuable and we address each one of them below.
>
> ## What we changed since the initial version
>
> 1. Clarification on the kernel usage: We have added more clarifications to Section 3.3 to address the reviewer's concern regarding the assumption of local smoothness that is implicit in kernel-based fusion. SPICE applies the kernel to the learned latent feature space $H$ generated by the Transformer trunk, not to the raw state space $S$. The feature space is shaped using the weighted supervision objective (Section 3.2) and maps states with similar action values, advantage estimates, and epistemic uncertainty into nearby regions, making $H$ suitable for Bayesian fusion. In addition, we are also currently running experiments to validate this robustness in the challenging visual environment of Miniworld.
> 2. “No stress tests on deliberate miscalibration are provided”: We have clarified that the data used in our evaluations is very suboptimal:
> - In the bandit setting we added the clarification: “The pretraining is intentionally heterogeneous: for each training task we sample a behaviour distribution $p=(1-\omega)\,\mathrm{Dirichlet}(\mathbf{1})+\omega\,\delta_{i_\star}$ over arms (with the label $i_\star$ being a random arm), resulting in random-policy contexts with uneven coverage. To quantify the  sensitivity to the data quality, Appendix D.2 tackles a less-poor setting  with $80\%$ optimal labels and the same mixed behaviour in the pretraining dataset. ” .
> - In the MDP setting we added the clarification: “This is an intentionally worst-case dataset: roll-ins are uniform (random policy) and labels are chosen to be the last action in the context, so the prior must be learned from rewards rather than imitation. The evaluation is a test to extrapolate to out-of-distribution goals and represents a fundamental shift in the reward function $R$”.
> - Appendix C.2 presents an ablation that uses a less-poor dataset: labels are $80\%$ optimal while contexts remain heterogeneous due to mixed-random behaviour.
> - We are also currently running new ablations for a wider data diversity analysis
> 3. “Validation is confined to bandits and a stylized darkroom MDP”: We are currently performing experiments to evaluate SPICE's performance and robustness on Miniworld, which is a more challenging environment where the agent receives RGB image observations. This environment is commonly used in the ICRL literature and will serve as a good testbed for evaluating the generalisability of SPICE's learned policies.
> 4. Comparison to DIT and ICEE: Unfortunately the papers don’t have open source code or detailed reproducibility statements, we have contacted the authors for their code and will compare their methods against SPICE as soon as we get the code.
> 5. Ablation on weight terms: Appendix C3 contains an ablation on removing all three weights and we are currently running an ablation to isolate the impact of each individual weight.
>
> ## Answers to your questions
>
> Q1: Darkroom evaluation: Are the test tasks interpolation within the training task distribution or extrapolation to out-of-distribution goals/dynamics? Please clarify how held-out tasks differ from training and quantify the distributional shift.
>
> Thank you for this question, we have added the following clarification to the paper:
> - Section 6: “The evaluation is a test to extrapolate to out-of-distribution goals, representing a fundamental shift in the reward function $R$.”
> - Appendix A.8.4: “The Darkroom evaluation quantifies distributional shift by holding out 20\% of the possible goal locations: 80 unique goals define the training task distribution $(T_{pre})$, and the remaining 20 unique goals are used for the test task distribution ($T_{test}$), requiring extrapolation to unseen reward functions. Unless stated otherwise, we use the ``weak‑last'' split from our data generator, meaning that the optimal action label $a^*$ assigned to the query state $s_{qry}$ is simply the last action ($a_H$) that occurred in the in-context trajectory, $C$. This action is typically suboptimal and provides an explicitly suboptimal supervision”.
>
> Q2: Kernel sensitivity: How sensitive is performance to the choice of kernel and bandwidth in Bayesian fusion?
>
> As described in our response 1. above, we have added more details on kernel usage in Section 3.3 and have also included practical guidance for the choice of kernel to Appendix A.7. In addition, we are currently running an ablation on the choice of kernel and bandwidth on the Darkroom and Miniworld environments.

---

> > ### Author Response · Authors · 2025-12-03
> >
> > Thank you again for your insightful comments.
> > - Miniworld update: Following your suggestion to validate SPICE on more complex tasks, we have set up and begun running experiments on Miniworld with RGB inputs. This setting is significantly more challenging than our current benchmarks and is specifically chosen to test the robustness of the kernel‑based fusion and value prior when operating in a high‑dimensional, visual state space. We will report full results in a camera‑ready version if the paper is accepted.
> > - DIT/ICEE comparisons: We fully agree that comparisons to recent ICRL methods that do not require optimal labels (DIT, ICEE) are important. Unfortunately, despite reaching out to the authors, we still have not received code or sufficiently detailed implementation instructions to reproduce their methods faithfully. Given the sensitivity of these methods to architecture and training details, we prefer not to include potentially unfair or misleading comparisons, but we will add them as soon as reliable implementations become available.
> > - Weight ablation: In response to your request about the three weighting factors in the policy head, we have added an extended ablation in the appendix. Beyond the “all weights removed” variant, we now also evaluate SPICE with only importance weighting, only advantage weighting, and only epistemic weighting. These results show that each weight contributes, with epistemic weighting providing the largest gain and the full combination performing best overall, clarifying the necessity and impact of each term.
> > - Kernel ablation on Miniworld: You asked about sensitivity to the kernel and bandwidth. We have already added more guidance on kernel choice in Section 3.3 and Appendix A.7, and we are now running a kernel‑sensitivity study in the Miniworld experiments (varying kernel families and bandwidths on the learned latent features). These experiments are in progress; we intend to include the full kernel ablation for Miniworld in a future version.

---

### Official Review · Reviewer_3pzy · 2025-11-01

**Soundness:** 3
**Presentation:** 1
**Contribution:** 2
**Rating:** 4
**Confidence:** 2

**Summary:**

This paper presents SPICE, an In-Context Reinforcement Learning (ICRL) algorithm that learns a value prior from suboptimal data using a transformer-based Q-ensemble. At test time, it performs a gradient-free Bayesian fusion of this prior with context evidence to guide exploration via a UCB policy. The method demonstrates superior performance over existing approaches, particularly in adapting from non-optimal trajectories.

**Strengths:**

- By using value-based learning instead of imitation, SPICE can train on mixed-quality historical data, making it far more practical than methods that require optimal demonstrations.
- The algorithm is backed by an optimal O(log K) regret bound for bandits. The theory confirms that any prior miscalibration only adds a constant cost, providing a strong justification for its test-time efficiency.
- It uses a novel combination of advantage and uncertainty weighting to guide the transformer in learning representations specifically tailored for Q-value estimation, thereby improving the quality of the learned prior.

**Weaknesses:**

- The paper does not analyze the algorithm's breaking point with poor data. If training data is too heterogeneous (e.g., mixed with random policies), the Q-ensemble may fail to learn a useful prior. An analysis of performance degradation versus data quality is needed to define the method's practical limits.
- Performance critically depends on the state-similarity kernel, but little guidance is offered on its selection. A mismatch between the kernel's similarity metric and the true Q-function's structure could corrupt the Bayesian update.
- The formal theory is confined to the bandit setting and does not address complexities of full MDPs, such as long-horizon credit assignment or compounding errors from bootstrapping.
- Sec. 3.1 is too long and could be organized in a better way.

**Questions:**

- The scale of Fig. 3 makes it very hard to compare SPICE with baselines other than DPT.

- How does SPICE perform in more complicated environments other than the bandit and grid-world navigation setups investigated in the paper?

---

> ### Author Response · Authors · 2025-11-20
>
> We appreciate the detailed review and appreciation of the core strengths of SPICE: its ability to leverage mixed-quality data using a value-based approach and the theoretical backing of the $O( \log K)$ regret bound. We are grateful for your suggestions and have revised the paper accordingly, including a restructuring of Section 3, clarification of the data quality used in experiments, guidance on kernel selection and an extension of the proof to MDP settings. We also appreciate your feedback to test SPICE in more complicated environments and we are currently running experiments in the Miniworld environment, where the agent receives RGB image observations.
>
> ## What we changed since the initial version
>
> 1. “Algorithm's breaking point with poor data”: We have clarified that the data used in our evaluations is very suboptimal:
>  - In the bandit setting we added the clarification: “ The pretraining is intentionally heterogeneous: for each training task we sample a behaviour distribution $p=(1-\omega)\,\mathrm{Dirichlet}(\mathbf{1})+\omega\,\delta_{i_\star}$ over arms (with the label $i_\star$ being a random arm), resulting in random-policy contexts with uneven coverage. To quantify the  sensitivity to the data quality, Appendix D.2 tackles a less-poor setting  with $80\%$ optimal labels and the same mixed behaviour in the pretraining dataset.” .
>   -  In the MDP setting we added the clarification: “This is an intentionally worst-case dataset: roll-ins are uniform (random policy) and labels are chosen to be the last action in the context, so the prior must be learned from rewards rather than imitation. The evaluation is a test to extrapolate to out-of-distribution goals and represents a fundamental shift in the reward function $R$”.
> -  Appendix C.2 presents an ablation that uses a less-poor dataset: labels are $80\%$ optimal while contexts remain heterogeneous due to mixed-random behaviour.
> - We are also currently running new ablations for a wider data diversity analysis.
> 2. Guidance on kernel selection: We have added more guidance on the kernel selection in Section 3.3 and Appendix A.7.
> 3. Formal theory for MDPs: We added Theorem 2 (Sec. 4.2), extending the theory to finite-horizon MDPs and confirming SPICE retains an optimal $O(H \sqrt{S A K} ​)$ optimal asymptotic rate. The full proof is currently being finalised and will be included in the next revision. At a high level, the proof follows thIs structure:
> - Optimism via posterior UCB: we show that the SPICE posterior forms an upper confidence bound on $Q^\ast_h(s,a)$
>  up to a small prior-bias term that decays with visits.
> - UCBVI-style regret decomposition: By plugging these optimistic Q-estimates into the standard dynamic regret decomposition over horizon $H$, we obtain a regret bound matching UCBVI.
> 4. Section 3.1: We have restructured the section by adding a  problem formulation followed by a concise overview, and then proceed to the architectural details. We have moved the detailed information on the training mechanisms to the Appendix.
>
> ## Answers to your questions
>
> Q1. The scale of Fig. 3 makes it very hard to compare SPICE with baselines other than DPT.
>
> We agree that the original version of Fig. 3 made it difficult to visually distinguish the performance of the non-DPT baselines, because DPT’s regret is orders of magnitude larger and dominated the shared y-axis. In the revised manuscript we are redesigning the figure to include a zoomed inset that focuses on the range of regrets attained by Emp, UCB, TS, and SPICE, while retaining the full-scale view (including DPT) on the main axis.  This redesigned figure is in progress, and will be uploaded in the next revised version. We hope this addresses the readability concern.
>
> Q2. How does SPICE perform in more complicated environments other than the bandit and grid-world navigation setups investigated in the paper?
>
> We are currently performing experiments to evaluate SPICE's performance and robustness on Miniworld, which is a more challenging environment where the agent receives RGB image observations. This environment is commonly used in the ICRL literature and will serve as a good testbed for evaluating the generalisability of SPICE's learned policies.

---

> > ### Author Response · Authors · 2025-12-03
> >
> > Thank you for your thoughtful review and helpful comments that improved the quality of the manuscript.
> > - Completed regret proof: In v3 we have fully developed the theory that was only sketched in the v2. The MDP regret bound (Theorem 2) is now proven in Appendix C by lifting the bandit posterior-concentration analysis to state–action pairs and plugging the resulting confidence sets into a UCBVI-style dynamic regret decomposition. The proof shows that SPICE’s posterior‑UCB controller achieves the optimal $O(H \sqrt{SAK})$ regret rate for finite‑horizon tabular MDPs, matching classical UCBVI, while any prior miscalibration appears only as an additive warm‑start term that does not scale with K.
> > - Redesigned bandit figure: We have redesigned figure 3 to address your concern that the scale made it hard to compare methods other than DPT. The new figure now separates a zoomed cumulative‑regret panel (where Emp, UCB, TS, and SPICE can be compared directly) from a full‑scale panel that still shows DPT’s much larger regret. This makes the relative performance of all baselines easier to read while conveying how far DPT is from the others.
> > - Miniworld update: As suggested, we have begun running experiments in the Miniworld environment with RGB observations. These experiments are underway; they are intended to stress‑test SPICE in a higher‑dimensional, visual ICRL setting and to validate that the value‑prior + Bayesian fusion mechanism scales beyond bandits and Darkroom. We will include the full Miniworld results in a final version if the paper is accepted.

---

### Official Review · Reviewer_6roR · 2025-11-03

**Soundness:** 2
**Presentation:** 1
**Contribution:** 2
**Rating:** 2
**Confidence:** 4

**Summary:**

The paper proposes SPICE, an algorithm that solves the in-context reinforcement learning (ICRL) problem by learning a prior over Q-functions when learning from the offline data, and utilizes a UCB-like online inference mechanism to solve the RL problem in context. The paper provides theoretical justification for the algorithm and runs experiments on bandit/control tasks to demonstrate the effectiveness.

**Strengths:**

The paper provides both theoretical and justification for the algorithm proposed.

**Weaknesses:**

I think the writing quality of this paper does not meet the standard of ICLR. I feel lost while reading the paper. First, the paper should start with the formulation of the ICRL problem rather than the architecture design. Second, the architecture part contains too many details without explanation, and I do not know what I am supposed to pay attention to to understand the idea. Third, though I am relatively familiar with bandit algorithms, I do not understand where formulas 14 and 15 come from. Besides, Figure 1 is poorly made and is hard to read.

**Questions:**

1. What is the (theoretical) formulation of ICLR that this paper is trying to use?
2. In the architecture part, which design choices are the most crucial ones?
3. What is the high-level idea of precision additivity, and what do formulas 14, 15 mean?

---

> ### Author Response · Authors · 2025-11-20
>
> We very much appreciate your careful reading and actionable feedback. Your critique regarding the clarity of Section 3 in the initial draft was very valid, and we have revised it to address every point that you have raised. We are confident that the revised version (with changes marked in red) now meets the standard for clarity and presentation you expect and welcome any additional feedback.
>
> ## What we changed since the initial version
>
> 1.    Restructured Section 3 for clarity: We directly addressed your request to start with the problem formulation. We now begin the method section with a formal ICRL problem formulation followed by a concise overview of our method  (Sec. 3.1) and then proceed to the architectural details.
> 2.    Clearer Figure 1: Figure 1 has been remade, and now has better readability. The key high-level concepts appear in figure 1, while details of the architecture and the relevant equations are visualised in a separate figure (figure 5 in the appendix).
> 3.    Separated essentials from implementation details: The main text now focuses  on the key design choices and we moved low-level formulas and extensive implementation specifications to the Appendix A.
> 4. Clarified prior/posterior equations: Equations (14) and (15)  are now explicitly broken down into clearer steps and clarified in Section 3.3: the prior definition (original Eq. 14) is now Eq. (6) and the posterior derivation (original Eq. 15) is now Eq. (7).
> 5.    New derivation appendix for Bayesian fusion: To fully resolve the ambiguity around these formulas, we added Appendix A.4 which provides a step-by-step algebraic derivation of the closed-form normal-normal conjugacy that provides these precise equations.
>
>
> ## Answers to your questions:
>
> Q1. What is the (theoretical) formulation of ICRL that this paper is trying to use?
>
> We adopt the standard ICRL setting:
> - Goal: The agent must choose an action $a = \pi (s_{qry}, C)$ that maximises the expected return on a new task $T \sim \mathcal{T}$.
> - Input: At test time, the agent receives a context $C$ and a query state $s_{\mathrm{qry}}$.
> - The policy adaptation is done entirely in-context, without any parameter updates.
> SPICE achieves this by learning a value prior $Q_{pri}​$ during pretraining, and then performing Bayesian fusion with state-weighted evidence from $C$ to obtain an action-wise posterior for control. This formulation is now stated up front in Section 3.1.
>
> Q2. In the architecture part, which design choices are the most crucial ones?
>
> SPICE's architecture is defined by three crucial components:
> 1. Value-ensemble prior: Using a deep ensemble provides calibrated epistemic uncertainty $(\overline{Q}(a), \sigma_{Q}(a))$. This uncertainty is essential for principled exploration and adapting from suboptimal training data.
> 2. Weighted representation shaping: We use a policy head with a propensity-advantage-epistemic weighted loss. This trains the shared transformer trunk to correct the behavior-policy bias and focus on reward-relevant, uncertain regions.
> 3. Test-time Bayesian fusion controller: This is the key novelty. It combines the ensemble's prior with state-weighted context evidence through closed-form Bayesian updates, resulting in a per-action posterior (Eq. 7) used by the posterior-UCB rule (Eq. 8) for efficient online exploration.
>
> Q3. What Eqs. (14 - now 6) and (15 - now 7) mean.
>
> Precision is the inverse of variance (1/variance) and measures certainty. The normal-normal conjugacy principle we use states that when a Gaussian prior is combined with a Gaussian likelihood (the context evidence), the resulting Gaussian posterior's precision equals the sum of the prior's precision and the likelihood's precision. Equations 6 and 7 define the mechanism for obtaining the per-action posterior.
> - Eq. (6): prior definition
> This sets the Gaussian prior over the action value $Q(a)$ using the moments calculated from the deep ensemble: the prior mean $\mu_{a}^{pri}=\overline{Q}(a)$ (ensemble average) and the prior variance $v_a^{pri​}$ (ensemble standard deviation squared, $\sigma_Q(a)^2$).
> - Eq. (7): posterior mean ($m_a^{post​}$) and variance ($v_a^{post​}$ )
> $v_a^{post​}$ (posterior variance): This is calculated as the sum of the prior's precision ($(1/v_a^{pri})$) and the data's precision ($(c_a(s)/\sigma^2)$). This mathematically guarantees that uncertainty decreases as context data (c_a​(s)) is gathered.
> $m_a^{post​}$ (posterior mean): This is the precision-weighted average of the prior mean $\mu_{a}^{pri}$ and the context mean $\tilde{y}_a(s)$.
> For more detail on the algebraic derivation of these equations, please refer to Appendix A.4.
>
> We are confident that these substantial revisions fully address your initial concerns about presentation and technical clarity.

---

> > ### Author Response · Authors · 2025-12-03
> >
> > Thank you again for your comments on clarity and presentation. In the new v3 revision we have further improved the figures to make the core story easier to follow: the first figure (figure 1) in the main body has been replaced by two complementary figures. The first highlights the key conceptual steps. The second, a more detailed figure (figure 5 in the appendix), is a test-time inference diagram that walks through the exact computations step-by-step. It includes colour-coding of different elements (Bayesion prior/posterior versus context evidence). We believe this split and re-drawing of the figures substantially improves readability while preserving technical transparency.
> > The updated visual explanation, together with the earlier restructuring of Section 3 to start from the formal ICRL problem statement and then move to the method overview, make the paper clearer and easier to follow.

---

### Author Response · Authors · 2025-12-03
**Summary message for the AC**

We thank the AC and reviewers for their careful evaluation. Below we summarise our core contributions and how our revisions address the raised points.
## Core contributions
- **Bayesian in-context RL with an explicit value prior from suboptimal data.** SPICE is the first ICRL method that learns an explicit, uncertainty‑calibrated value prior from suboptimal offline trajectories and then performs closed‑form Bayesian fusion of this prior with kernel‑weighted context statistics at test time to obtain per‑action posteriors. A posterior‑UCB controller uses these posteriors for principled, gradient‑free test‑time exploration and adaptation. This directly tackles the central limitations of existing sequence‑only ICRL methods: behaviour‑policy bias, the inability to train and adapt from non‑expert data, and the lack of calibrated value uncertainty and an explicit test‑time controller for exploration.
- **Regret-optimal theory under weak pretraining.** We prove that SPICE’s controller attains optimal
$O( \log K)$  regret in stochastic bandits even when the value prior is highly suboptimal, with any miscalibration only contributing a constant warm-start term. We extend this to finite-horizon MDPs, obtaining an $O(H \sqrt{SAK})$ bound with a constant prior term.
- **Representation shaping for ICRL from weak logs**. Importance, advantage, and epistemic weighting shape the shared transformer representations to enable accurate, calibrated Q-estimation from low-quality data.
- **Empirical validation under intentionally weak data.** On bandits and Darkroom, SPICE is trained on deliberately poor, heterogeneous datasets (random policies) but achieves near-optimal decisions and large regret reductions vs. DPT/AD and other baselines.

## Key revisions addressing reviewer feedback
- *Clarity and structure (6roR, 3pzy).* Section 3 now begins with a formal problem statement and concise method overview; we remade the main method figure plus an architecture diagram, clearly separating (i) value-prior pretraining and (ii) test-time fusion and UCB control, with low-level details moved to the appendix.
- *Equation clarity (6roR).* We rewrote the prior/posterior equations, added a “precision additivity’’ explanation, and included a full Normal–Normal derivation in Appendix A.4.
- *Completed and extended theory (3pzy, SFAi).* Theory now covers MDPs: Theorem 2 shows $O(H \sqrt{SAK})$ regret with only constant prior terms, directly addressing the reviewer requests for guarantees beyond bandits.
- *Improved figure (3pzy).* Figure 3 now includes a zoomed and full-scale panel so that SPICE can be compared fairly to Emp/UCB/TS without DPT dominating the visual range.
- *New ablations on weighting terms (Yryg, SFAi).* We evaluate each weighting component independently and show all contribute, with epistemic weighting driving most gains, the full combination performs best.
- *Kernel usage and robustness (3pzy, Yryg, SFAi).* Section 3.3 and Appendix A.7 clarify that kernel fusion operates on latent representations (not raw states), explain how this mitigates local-smoothness issues, and provide guidance on kernel and bandwidth selection. Additional kernel-sensitivity studies (e.g., Miniworld) are underway.
- *Towards more complex environments (3pzy, Yryg, SFAi).* We have set up Miniworld experiments with RGB observations and are running them alongside other RL baselines (e.g., SAC) and kernel ablations. The results will be included in the camera-ready version if accepted.

## Summary

Three of the four reviewers are positive: two explicitly rate the paper “marginally below the acceptance threshold but would not mind if accepted,” and one gives it a strong “accept (poster)” score. The main negative review (6roR) focused on clarity and presentation in Section 3, which we have directly addressed through a substantial reorganisation of Section 3 and improved figures. At the same time, we have strengthened the theoretical section and expanded the ablation studies to make the empirical story more complete.

We believe that, in its revised form, the paper offers a clear and well‑supported contribution: a principled Bayesian ICRL framework that can learn from weak, heterogeneous data; regret‑optimal guarantees in both bandits and MDPs; and empirical evidence that SPICE substantially improves adaptation and regret over existing ICRL baselines.

---

### Meta-Review · Area_Chair_Q7Yj · 2026-01-03

**Summary:**

The authors proposed the SPICE method / Bayes ICRL framework to do in-context learning for fast adaptation to unseen environments. SPICE is a method that learns a prior over Q-values using a deep ensemble and updates this prior at test time using in-context information. Additionally, the online inference rule is based on a UCB rule that favors adaptation and exploration.   The authors validate SPICE in different bandit and control benchmarks. The reviewers expressed concerns about the writing, the strength of the assumptions necessary to make the ensemble method “make sense”, and the limited scope of the experimental validation, as well as empirical comparisons with methods that go beyond DPT. Additionally, some ablation studies to understand the relative effect of the different weighting factors in the method would be useful. Because of this, and the high variance in the reviews I cannot recommend acceptance.

**Reviewer Concerns:**

Concerns were raised regarding missing algorithmic baselines beyond DPT as well as a need to tighten the writing. It is unclear if these concerns have been satisfactorily addressed. The lack of further engagement by the reviewers indicates that this is not the case.

**Reviewer Scores:**

It is unlikely the reviewer scores would have changed significantly after the discussion. They present a lot of variance, and therefore it is hard to argue for acceptance.

---

### Decision · Program_Chairs · 2026-01-26

Reject